# Vicinity-Guided Discriminative Latent Diffusion for Privacy-Preserving Domain Adaptation

**Jing Wang**[1]    **Wonho Bae**[1]    **Jiahong Chen**[3]    **Wenxu Wang**[4]    **Junhyug Noh**[2]

[1]University of British Columbia    [2]Ewha Womans University
[3]Amazon Inc.*    [4]Ocean University of China
j.wang94@alumni.ubc.ca    junhyug@ewha.ac.kr

## Abstract

Recent work on latent diffusion models (LDMs) has focused almost exclusively on generative tasks, leaving their potential for discriminative transfer largely unexplored. We introduce Discriminative Vicinity Diffusion (DVD), a novel LDM-based framework for a more practical variant of source-free domain adaptation (SFDA): the source provider may share not only a pre-trained classifier but also an auxiliary latent diffusion module, trained once on the source data and never exposing raw source samples. DVD encodes each source feature's label information into its latent vicinity by fitting a Gaussian prior over its $k$-nearest neighbors and training the diffusion network to "drift" noisy samples back to label-consistent representations. During adaptation, we sample from each target feature's latent vicinity, apply the frozen diffusion module to generate source-like cues, and use a simple InfoNCE loss to align the target encoder to these cues, explicitly transferring decision boundaries without source access. Across standard SFDA benchmarks, DVD outperforms state-of-the-art methods. We further show that the same latent diffusion module enhances the source classifier's accuracy on in-domain data and boosts performance in supervised classification and domain generalization experiments. DVD thus reinterprets LDMs as practical, privacy-preserving bridges for *explicit* knowledge transfer, addressing a core challenge in source-free domain adaptation that prior methods have yet to solve. Code is available on our Github: https://github.com/JingWang18/DVD-SFDA.

## 1   Introduction

Latent Diffusion Models (LDMs) have recently gained prominence in artificial intelligence for executing the diffusion process in a lower-dimensional latent space [Rombach et al., 2022]. Compared to pixel-level diffusion models [Ho et al., 2020, Song et al., 2021], LDMs offer notable improvements in efficiency by removing the need for computationally expensive gradient updates at every diffusion step [Dhariwal and Nichol, 2021]. Their flexibility in incorporating multi-modal signals, such as text prompts for image generation [Avrahami et al., 2023], motivates us to investigate whether LDMs can be repurposed for a *discriminative* goal, specifically for *source-free domain adaptation* (SFDA), where a core challenge remains unsolved: enabling *explicit knowledge transfer* from a source model without access to source data.

In SFDA [Liang et al., 2020], a classifier trained on a labeled source domain must adapt to an unlabeled target domain without directly accessing the source data. This setting arises in scenarios where original training data cannot be shared due to privacy or proprietary restrictions (*e.g*., medical images from different hospitals). Even if we can release the source model parameters, we typically

---

*Work conducted outside Amazon.

39th Conference on Neural Information Processing Systems (NeurIPS 2025).

cannot provide the raw source data to the practitioners. Existing SFDA methods struggle to deliver *explicit* knowledge transfer under such constraints. This motivates our reformulation of SFDA as *privacy-preserving domain adaptation*, where the source provider may release a lightweight auxiliary module, without exposing raw data, to assist downstream adaptation. Our central question is: *Can LDMs enable explicit discriminative knowledge transfer under these constraints?* We address this by proposing **Discriminative Vicinity Diffusion** (**DVD**).

DVD builds on the smoothness assumption from semi-supervised learning [Iscen et al., 2019], positing that nearby points in feature space tend to share the same labels. Concretely, DVD encodes each source data's label information into its *latent vicinity* (the $k$-nearest neighbors in latent space) by defining a Gaussian prior around this vicinity in the diffusion process. Intuitively, the learned drift-only diffusion then acts like reverse diffusion, pulling noisy latent samples back onto the source data manifold defined by that vicinity and restoring label-consistent representations. Leveraging the property of latent diffusion models, which iteratively guide samples toward more probable regions in latent space, DVD can generate source-like latent samples for any new feature whose encoded representation lies near a learned vicinity. Thus, during target adaptation, DVD uses the target data's *own* local neighbors to generate source-like features, explicitly transferring source decision boundaries without exposing raw source data.

While typical SFDA setups allow the *source provider* to share only a backbone classifier (*e.g.*, ResNet [He et al., 2016]), we extend this by permitting the release of an additional DVD module, without raw data, thus framing the problem as privacy-preserving domain adaptation. Although this introduces minimal overhead, it enables the target user not only to adapt under strict privacy constraints but also to better understand the adaptation process. The source provider also benefits: DVD augments latent features during training, yielding more robust decision boundaries and improved source-domain accuracy. Beyond SFDA, we show that DVD further enhances *domain generalization*, improving performance on unseen domains without adaptation. Our main contributions are:

- We introduce *Discriminative Vicinity Diffusion* (DVD), an LDM-based framework for explicit cross-domain knowledge transfer without sharing source data.
- We propose a *latent vicinity guidance* mechanism that leverages latent $k$-nearest neighbors ($k$-NNs) to parameterize Gaussian priors more effectively than simple noise injection in discriminative tasks.
- We demonstrate that DVD achieves state-of-the-art results on multiple SFDA benchmarks by surpassing existing methods.
- We further show that DVD improves source-domain accuracy through latent augmentation and exhibits strong domain generalization on unseen domains.

## 2 Related Work

**Contrastive Source-Free Domain Adaptation.** The motivation behind SFDA is to enable domain adaptation when sharing source data is restricted due to privacy, security, or storage constraints. In SFDA, we adapt a source-trained model to an unlabeled target domain without access to raw source data [Liang et al., 2020]. Contrastive methods apply an InfoNCE-style loss on model outputs to cluster target embeddings by class predictions [Chen et al., 2020, Yang et al., 2022, Zhang et al., 2022], offering strong performance with minimal complexity. We adopt this simple contrastive objective to align target features with source-derived cues.

**Diffusion and Latent Diffusion Models.** Diffusion generative models iteratively denoise samples via stochastic differential equations (SDEs), often incurring high computational cost [Ho et al., 2020, Song et al., 2021]. Latent diffusion models (LDMs) alleviate this by operating in a low-dimensional latent space, significantly reducing inference time and resource usage [Schramowski et al., 2023, Avrahami et al., 2023]. This efficiency underpins our choice to repurpose LDMs for discriminative and source-free transfer rather than generation alone.

**Diffusion for Domain Adaptation.** Prior work has leveraged diffusion models for data augmentation to enhance domain adaptation performance, *e.g.*, text-to-image diffusion to synthesize target-style samples with source labels [Benigmim et al., 2023] or 3D generative diffusion for shape transfer [Kim and Chun, 2023]. These methods operate in pixel space and focus on expanding training data, whereas DVD applies latent diffusion to directly transfer discriminative cues via local Gaussian priors over $k$-nearest neighbors, enabling explicit source-free knowledge transfer.

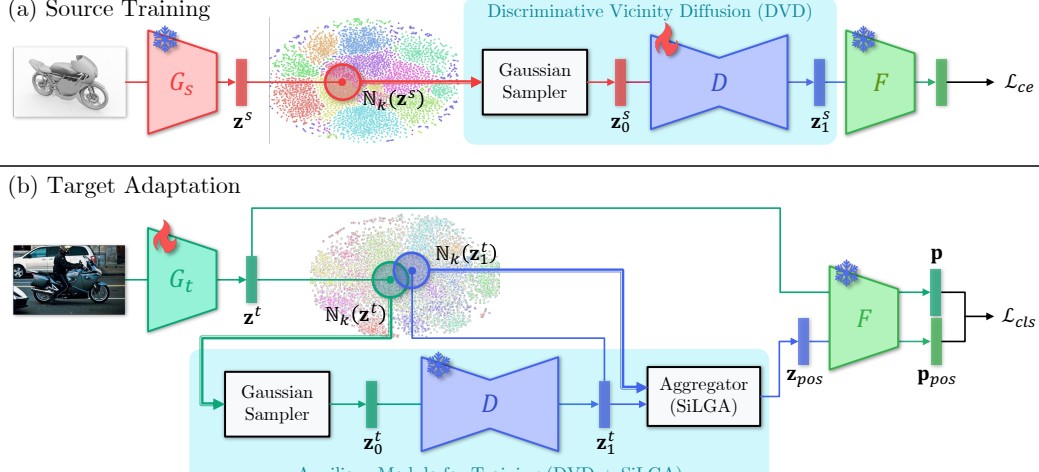

Figure 1: **Framework overview.** In source pre-training, our DVD aligns source features for consistent predictions within their vicinities. In target adaptation, the target latent vicinity guides DVD to generate features from nearby source latent vicinities based on their similarities. The parameterization of diffusion priors enables this guidance across domains. After these training phases, inference requires only the classifier $F \circ G_t$, without invoking $D$.

**Generative SFDA Methods.** Many SFDA approaches generate source-like features or prototypes without access to raw data: CPGA builds class prototypes via contrastive learning [Qiu et al., 2021], 3C-GAN uses a GAN to produce pseudo-labeled target-style images [Li et al., 2020], and SFADA synthesizes proxy samples [He et al., 2024]. In contrast, DVD employs a deterministic and latent-space diffusion process guided by label smoothness in local vicinities, sidestepping GAN training and complex data density modeling in the target domain.

## 3 Method

We introduce *Discriminative Vicinity Diffusion* (DVD), a latent-space diffusion framework for Privacy-Preserving Domain Adaptation (a variant of source-free domain adaptation). DVD operates with a *drift-only* diffusion model [Heitz et al., 2023] in the encoder's latent space, enabling efficient encoding and retrieval of source-domain discriminative knowledge without exposing raw source data.

### 3.1 Preliminaries

**Notation and Setup.** Let $\mathbf{S} = \{(\mathbf{x}^{s,i}, \mathbf{y}^{s,i})\}_{i=1}^{N}$ be the labeled source domain and $\mathbf{T} = \{\mathbf{x}^{t,i}\}_{i=1}^{M}$ the unlabeled target domain. We denote the classifier by $f = F \circ G$, where $G$ (*e.g.*, ResNet [He et al., 2016]) maps inputs into a latent space and $F$ is a small fully-connected head.

*A Variant of SFDA:* In typical SFDA, only a pre-trained classifier $(G_s, F)$ is shared. We generalize this to a privacy-preserving domain adaptation setting, where the source provider can also release an auxiliary module without ever exposing raw data. Specifically, we introduce a latent diffusion module $D$ trained once on $\mathbf{S}$. By encoding source discriminative cues into a compact and privacy-preserving representation, $D$ enables more flexible and robust adaptation to $\mathbf{T}$ under strict privacy constraints.

**Latent Vicinities.** Both at source pre-training and at target adaptation time, we rely on *k-nearest neighbors* ($k$-NNs) in latent space to capture local geometry. Specifically, for the source side, we maintain a *feature bank*:

$$\mathbb{B}_{\mathbf{S}} = \left\{ G_s(\mathbf{x}^{s,j}) \mid \mathbf{x}^{s,j} \in \mathbf{S} \right\}, \quad j = 1, \dots, N. \tag{1}$$

Similar to [Huang et al., 2019], given a query's feature vector $\mathbf{z}^{s,i} = G_s(\mathbf{x}^{s,i})$, we use the cosine similarity to determine its latent $k$-NNs within $\mathbb{B}_{\mathbf{S}}$:

$$d(\mathbf{z}^{s,i}, \mathbb{B}_{\mathbf{S}}^j) := \frac{\mathbf{z}^{s,i} \cdot \mathbb{B}_{\mathbf{S}}^j}{||\mathbf{z}^{s,i}|| \, ||\mathbb{B}_{\mathbf{S}}^j||}, \tag{2}$$

$\mathbb{B}_{\mathbf{S}}^j$ denotes the $j$-th element of $\mathbb{B}_{\mathbf{S}}$ for input data $\mathbf{x}^{s,j}$. Distinct notations are used for the query's encoded features $\mathbf{z}^{s,i}$ and features in the bank $\mathbb{B}_{\mathbf{S}}^j$, which reflects potential differences before the bank is updated with the current encoder parameters. Thus, the latent vicinity comprising latent $k$-NNs of the query $\mathbf{z}^{s,i}$ is defined as:

$$\mathbb{N}_k(\mathbf{z}^{s,i}) := \arg\max{}_{\max(\mathcal{K}) \leq N, |\mathcal{K}| = k} \sum_{j \in \mathcal{K}} d(\mathbf{z}^{s,i}, \mathbb{B}_{\mathbf{S}}^j), \tag{3}$$

where $\mathcal{K}$ is a set that includes indices of all $k$-NN features.

An analogous bank $\mathbb{B}_{\mathbf{T}}$ and procedures apply to target-domain features at adaptation time.

**Diffusion Background.** Conventional diffusion generative models define a forward noising process and learn to reverse it by using a score function that estimates the gradient of the log-likelihood of the transition probability distribution in the SDE. While effective, pixel-level diffusion typically requires (i) large computational resources per step and (ii) explicit modeling of a noise schedule $\sigma_t(\cdot)$. In contrast, we leverage a *deterministic* (drift-only) method in the latent feature space to implement DVD, eliminating the need for stochastic noise estimation and repeated encoder backpropagation.

## 3.2 Deterministic Latent Diffusion

**Blending and Drift.** Suppose we want to transform a latent vector from one distribution $\mathbf{z}_0 \sim p_0$ into another $\mathbf{z}_1 \sim p_1$. We define a blending operation with a scalar $\alpha \in [0, 1]$:

$$\mathbf{z}_\alpha = (1 - \alpha)\mathbf{z}_0 + \alpha\mathbf{z}_1. \tag{4}$$

A *drift* function $D(\mathbf{z}_\alpha, \alpha)$, parameterized by a neural network, then refines the blended state by predicting the vector difference $(\mathbf{z}_1 - \mathbf{z}_0)$. Discretizing $\alpha$ into $T$ steps,

$$\alpha_0 = 0, \; \alpha_1 = \tfrac{1}{T}, \; \ldots, \; \alpha_T = 1,$$

we iteratively update

$$\mathbf{z}_{\alpha_{t+1}} = \mathbf{z}_{\alpha_t} + (\alpha_{t+1} - \alpha_t) D(\mathbf{z}_{\alpha_t}, \alpha_t), \tag{5}$$

starting with $\mathbf{z}_{\alpha_0} = \mathbf{z}_0$. After $T$ steps, we obtain $\mathbf{z}_{\alpha_T} \approx \mathbf{z}_1$. Following [Heitz et al., 2023], our drift network $D(\mathbf{z}_{\alpha_t}, \alpha_t)$ is trained to predict the global difference $(\mathbf{z}_1 - \mathbf{z}_0)$ for each pair, rather than a local increment at each step. This formulation ensures that the drift field always points from the start $\mathbf{z}_0$ to the end $\mathbf{z}_1$, while the actual local increment is scaled by $(\alpha_{t+1} - \alpha_t)$. As a result, this deterministic update path avoids error accumulation and guarantees transport along the straight line between $\mathbf{z}_0$ and $\mathbf{z}_1$. The detailed rationale for this formulation is provided in Appendix A.1.

Although this update proceeds in $T$ discrete steps, it does not constitute an iterative optimization loop, instead, it is a single, deterministic drift path. Empirically, we find that a small $T$ (*e.g.*, 8) suffices to recover label-consistent features, thanks to the strong regularization provided by the latent vicinity prior and the pre-trained classifier. Unlike residual regression or loss-minimizing iterations, these updates follow a pre-defined transport path that is explicitly regularized by the Gaussian prior of the local latent vicinity. Each intermediate state therefore remains within a label-consistent region, preventing arbitrary drift or error accumulation. Moreover, Appendices B.5 and B.6 demonstrate that DVD is robust across vicinity radii and drift steps, confirming the stability of the drift mechanism.

**Training the Drift Function.** To train the drift function $D$, we sample pairs $(\mathbf{z}_0, \mathbf{z}_1) \sim (p_0 \times p_1)$, randomly pick $\alpha \in [0, 1]$, and form $\mathbf{z}_\alpha$ via (4). Our training objective is to minimize:

$$\mathcal{L}_{dif} = \mathbb{E}_{\alpha_t, \mathbf{z}_{\alpha_t}} [||D(\mathbf{z}_{\alpha_t}, \alpha_t) - \mathbb{E}_{(\mathbf{z}_0, \mathbf{z}_1)|(\mathbf{z}_{\alpha_t}, \alpha_t)}[\mathbf{z}_1 - \mathbf{z}_0]||^2]. \tag{6}$$

This encourages $D$ to reproduce the average difference between the original $\mathbf{z}_0$ and $\mathbf{z}_1$. If $\mathbf{z}_1$ corresponds to a labeled source feature, we additionally impose a cross-entropy term to ensure the final state $\mathbf{z}_{\alpha_T}$ (*i.e.,* $\hat{\mathbf{z}}_1$) is classified correctly by $F$, where $\sigma(\cdot)$ denotes the softmax function:

$$\mathcal{L}_{\text{ce}} = -\frac{1}{N} \sum_{i=1}^{N} \log \sigma\big(F(\hat{\mathbf{z}}_1^i)\big)_{\mathbf{y}^{s,i}}. \tag{7}$$

Algorithms 1 and 2 detail the training and sampling procedures. Notably, *no* stochastic noise is added each step, and the forward pass does not require computing encoder gradients, making it computationally lighter than pixel-level diffusion and well-suited for discriminative purpose.

### 3.3 Storing Source Knowledge via DVD

We begin by training a *source classification model* $f_s = F \circ G_s$ on the labeled source domain $\mathbf{S}$ using cross-entropy. Once $f_s$ converges, we *freeze $G_s$ and $F$* and focus on training the drift function $D$. The goal is to map each local latent prior surrounding a source embedding back to that embedding itself, thus "storing" discriminative knowledge in $D$ without the need to revisit source data.

Concretely, for each source pair $(\mathbf{x}^{s,i}, \mathbf{y}^{s,i})$, we first compute its latent representation $\mathbf{z}^{s,i} = G_s(\mathbf{x}^{s,i})$. We then identify the latent vicinity $\mathbb{N}_k(\mathbf{z}^{s,i})$ for each source data using $k$-NNs in latent space and compute its mean and variance:

$$\mu_0^s = \frac{1}{k} \sum_{j=1}^{k} \mathbb{N}_k^j(\mathbf{z}^{s,i}), \quad \sigma_0^{s2} = \frac{1}{k} \sum_{j=1}^{k} \Big(\mathbb{N}_k^j(\mathbf{z}^{s,i}) - \mu_0^s\Big)^2. \tag{8}$$

These define a Gaussian prior $p_0^s = \mathcal{N}(\mu_0^s, \sigma_0^{s2})$; we sample $\mathbf{z}_0^s \sim p_0^s$ as the "start" state and set $\mathbf{z}_1^s = \mathbf{z}^{s,i}$ as the "target" state. Finally, we apply Algorithm 1 to train $D$ so that drifting from $\mathbf{z}_0^s$ to $\mathbf{z}_1^s$ under $D$ yields a representation that $F$ correctly classifies as $\mathbf{y}^{s,i}$.

This teaches $D$ to "pull" noisy latent samples onto the true source manifold, effectively encoding all source discriminative cues within $D$ and eliminating any need to access $\mathbf{S}$ during adaptation. Moreover, these latent-space augmentations also improve the source-domain classifier's robustness, further benefiting the source provider.

---

**Algorithm 1:** Latent Diffusion Training

**Input**:
  $(\mathbf{z}_0, \mathbf{z}_1) \sim (p_0 \times p_1)$
  $\alpha \sim \text{Uniform}[0, 1]$
  $D$'s parameters $\phi$; steps $T$

**for** $t = 0$ **to** $T$ **do**
  |  $\mathbf{z}_{\alpha_t} \leftarrow (1 - \alpha_t)\mathbf{z}_0 + \alpha_t \mathbf{z}_1$;
  |  Update $\phi$ using $\mathcal{L}_{\text{dif}}$ (Eq. 6);
  |  Sample $\hat{\mathbf{z}}_1$ via Algorithm 2;
  |  If $\mathbf{z}_1$ has label, minimize $\mathcal{L}_{\text{ce}}$ (Eq. 7);
**end**

**Output**: Updated parameters $\phi$

---

**Algorithm 2:** Latent Diffusion Sampling

**Input**:
  $(\mathbf{z}_0, \mathbf{z}_1) \sim (p_0 \times p_1)$
  Discrete $\alpha_t = \frac{t}{T}$; steps $T$

**Initialize:** $\mathbf{z}_{\alpha_0} \leftarrow \mathbf{z}_0$
**for** $t = 0$ **to** $T - 1$ **do**
  |  $\mathbf{z}_{\alpha_{t+1}} \leftarrow \mathbf{z}_{\alpha_t} + (\alpha_{t+1} - \alpha_t) D(\mathbf{z}_{\alpha_t}, \alpha_t)$;
**end**

**Output**: $\hat{\mathbf{z}}_1 = \mathbf{z}_{\alpha_T} \approx \mathbf{z}_1$

---

### 3.4 Target Adaptation

Given only unlabeled target data $\mathbf{T}$ and pre-trained $(G_s, F, D)$, we initialize a new encoder $G_t$ from $G_s$ and fine-tune *only* $G_t$, keeping $F$ and $D$ frozen. During adaptation, the diffusion module $D$ generates source-aligned cues that guide $G_t$ to shift its latent outputs onto the source manifold, ensuring they lie within the frozen classifier $F$'s decision boundaries. At inference, $D$ is no longer used with the predictions made by $F \circ G_t$.

**Local Prior Sampling for Target Data.** Given a target sample $\mathbf{x}^{t,i}$, we encode it as $\mathbf{z}^{t,i} = G_t(\mathbf{x}^{t,i})$. We then identify its $k$-NNs in the target feature bank $\mathbb{B}_{\mathbf{T}}$ and compute their mean and variance. This defines a Gaussian prior $p_0^t = \mathcal{N}(\mu_0^t, \sigma_0^{t2})$. Sampling $\mathbf{z}_0^t \sim p_0^t$ and applying Algorithm 2 with the frozen drift model $D$ produces $\mathbf{z}_1^t$, which carries source-aligned, discriminative cues tailored to $\mathbf{z}^{t,i}$. Importantly, DVD does not collapse the entire target distribution onto the source; rather, it guides each target feature toward its most relevant source-informed latent cluster, ensuring semantic alignment with the frozen source classifier's decision boundaries.

**Source-Informed Latent Geometry Aggregation (SiLGA).** Although $\mathbf{z}_1^t$ carries source cues, relying on it alone can lead to label mismatches if the domain shift is large. To mitigate this, we blend

$\mathbf{z}_1^t$ with local target neighbors:

$$\mathbf{z}_{pos}^i \ := \ \frac{\mathbf{z}_1^{t,i} \ + \ \sum_{\mathbf{z}\in\mathbb{N}_k(\mathbf{z}_1^{t,i})}\mathbf{z}}{k+1}, \tag{9}$$

where $\mathbf{z}\in\mathbb{N}_k(\mathbf{z}_1^{t,i})$ are the top-$k$ neighbors drawn from the original target feature bank $\mathbb{B}_\mathbf{T}$, recalculated at each update rather than taken from the generated features. This ensures that the aggregation consistently reflects the current latent geometry of the target distribution. After sampling from the nearest source-informed Gaussian prior, SiLGA blends this feature with the target's local $k$-NN centroid, which is especially beneficial under large domain gaps (*e.g.,* VisDA-C) where target features are far from source centroids.

**Contrastive Clustering.**    We then adopt an InfoNCE-style contrastive objective [Oord et al., 2018, Chen et al., 2020, Wang et al., 2024] to cluster unlabeled targets. Concretely, let

$$p^i = \sigma\big(F(\mathbf{z}^{t,i})\big), \quad p_{pos}^i = \sigma\big(F(\mathbf{z}_{pos}^i)\big), \tag{10}$$

where $\sigma(\cdot)$ denotes a softmax function and $F$ is frozen. For a mini-batch of size $m$, we minimize

$$\mathcal{L}_{cls} \ = \ -\sum_{i=1}^m \log\Big[\frac{\exp\big(p^i\cdot p_{pos}^i \,/\, \tau\big)}{\sum_{j\neq i}\exp\big(p^i\cdot p^j/\tau\big)}\Big], \tag{11}$$

where $\tau$ is a temperature hyperparameter. To maintain focus on exploring the role of DVD in knowledge transfer without source data, we avoid incorporating additional techniques, such as momentum updates [He et al., 2020]. Minimizing $\mathcal{L}_{cls}$ pulls each target feature $\mathbf{z}^{t,i}$ toward its SiLGA-based positive $\mathbf{z}_{pos}^i$, effectively aligning $G_t$ with the source-discriminative cues learned in $D$. After sufficient epochs, $G_t$ converges to a representation that classifies $\mathbf{T}$ accurately without seeing any source data, thus satisfying SFDA constraints. At inference time, only $G_t$ and $F$ are used, where $D$ is not invoked, so there is no additional computational overhead beyond a standard forward pass.

## 4   Experiments

We evaluate DVD on three tasks: source-free domain adaptation (Section 4.1), supervised single-domain classification (Section 4.2), and domain generalization (Section 4.3). We then analyze hyperparameter sensitivity (Section 4.4) and runtime efficiency (Section 4.5). Additional results and ablation studies are provided in the Appendix.

### 4.1   Source-Free Domain Adaptation

In this section, we evaluate DVD's effectiveness in SFDA, focusing on its role in discriminative knowledge transfer without source data during target adaptation. We extend our discussion on DVD's relaxed SFDA protocol, where an auxiliary module is trained once offline while preserving strict source-free privacy during adaptation, and provide additional comparisons with conventional domain adaptation methods in Appendix C.1.

**Datasets.**    We conduct experiments on three widely recognized SFDA benchmarks:

- **Office-31** [Saenko et al., 2010]: 4,652 images across 31 classes collected from three domains – Amazon (**A**), Webcam (**W**), and DSLR (**D**).
- **Office-Home** [Venkateswara et al., 2017]: 15,500 images in 65 classes from four domains – Artistic (**Ar**), Clipart (**Cl**), Product (**Pr**), and Real-World (**Rw**).
- **VisDA-C 2017** [Peng et al., 2017]: 280,000 images spanning 12 classes, where the source domain is rendered via 3D models, and the target domain consists of real images captured by RGB cameras.

**Experimental Setup.**    To simplify the implementation of DVD, we adopt a consistent framework for all benchmarks. ResNet-50 serves as the encoder $G$ for Office-31 and Office-Home, while ResNet-101 is used for VisDA-C 2017 to align with existing SFDA baselines. A two-layer linear head $F$ performs classification. We use a conditional UNet [Ho et al., 2020] for $D$, with 16 diffusion steps in both training and inference. An SGD optimizer (learning rate $3\times10^{-3}$, momentum 0.9, batch size 128) is used for parameter updates. We employ the InfoNCE objective from SimCLR [Chen et al., 2020]

Table 1: Comparison of the SFDA methods on *VisDA-C 2017* (ResNet-101).

| Method | Add. | plane | bcycl | bus | car | horse | knife | mcycl | person | plant | sktbrd | train | truck | Avg. ± s.d. |
|---|---|---|---|---|---|---|---|---|---|---|---|---|---|---|
| ResNet-101 [He et al., 2016] | | 55.1 | 53.3 | 61.9 | 59.1 | 80.6 | 17.9 | 79.7 | 31.2 | 81.0 | 26.5 | 73.5 | 8.5 | 52.4 |
| SHOT [Liang et al., 2020] | | 94.3 | 88.5 | 80.1 | 57.3 | 93.1 | 94.9 | 80.7 | 80.3 | 91.5 | 89.1 | 86.3 | 58.2 | 82.9 |
| HCL [Huang et al., 2021] | | 93.3 | 85.4 | 80.7 | 68.5 | 91.0 | 88.1 | 86.0 | 78.6 | 86.6 | 88.8 | 80.0 | **74.7** | 83.5 |
| DaC [Zhang et al., 2022] | | 96.6 | 86.8 | **86.4** | 78.4 | 96.4 | 96.2 | **93.6** | **83.8** | 96.8 | 95.1 | 89.6 | 50.0 | 87.3 |
| NRC++ [Yang et al., 2023b] | | 96.8 | 91.9 | 88.2 | 82.8 | 97.1 | 96.2 | 90.0 | 81.1 | 95.2 | 93.8 | 91.1 | 49.6 | 87.8 |
| SFADA [He et al., 2024] | | 94.2 | 79.6 | 79.8 | 65.7 | 92.6 | 94.1 | 87.3 | 80.8 | 88.1 | 91.4 | 83.3 | 55.0 | 82.7 |
| CPD [Zhou et al., 2024] | | 96.7 | 88.5 | 79.6 | 69.0 | 95.9 | 96.3 | 87.3 | 83.3 | 94.4 | 92.9 | 87.0 | 58.7 | 85.8 |
| 3C-GAN [Li et al., 2020] | ✓ | 95.4 | 75.8 | 70.5 | 73.9 | 92.4 | 94.3 | 89.4 | 82.5 | 91.7 | 90.1 | 85.2 | 50.3 | 82.6 |
| SFADA [He et al., 2024] | ✓ | 94.9 | 80.4 | 80.6 | 68.2 | 94.3 | 94.0 | 86.5 | 82.1 | 91.4 | 92.2 | 85.0 | 51.4 | 83.4 |
| DVD (Ours) | ✓ | **98.4** | **92.1** | 83.9 | **83.6** | **98.1** | **96.5** | 92.1 | 82.9 | **97.0** | **95.2** | **92.6** | 54.6 | **88.9 ± 0.6** |

Table 2: Comparison of the SFDA methods on *Office-Home* (ResNet-50).

| Method | Add. | Ar → | | | Cl → | | | Pr → | | | Rw → | | | Avg. ± s.d. |
|---|---|---|---|---|---|---|---|---|---|---|---|---|---|---|
| | | Cl | Pr | Rw | Ar | Pr | Rw | Ar | Cl | Rw | Ar | Cl | Pr | |
| ResNet-50 [He et al., 2016] | | 34.9 | 50.0 | 58.0 | 37.4 | 41.9 | 46.2 | 38.5 | 31.2 | 60.4 | 53.9 | 41.2 | 59.9 | 46.1 |
| DaC [Zhang et al., 2022] | | 59.5 | 79.5 | 81.2 | **69.3** | 78.9 | 79.2 | 67.4 | 56.4 | 82.4 | **74.0** | **61.4** | 84.4 | 72.8 |
| NRC++ [Yang et al., 2023b] | | 57.8 | **80.4** | 81.6 | 69.0 | 80.3 | 79.5 | 65.6 | 57.0 | 83.2 | 72.3 | 59.6 | 85.7 | 72.5 |
| SFADA [He et al., 2024] | | 56.1 | 78.0 | 81.6 | 68.5 | 79.5 | 78.5 | 67.8 | 56.0 | 82.3 | 73.6 | 57.8 | 83.0 | 71.9 |
| CPD [Zhou et al., 2024] | | 59.1 | 79.0 | 82.4 | 68.5 | 79.7 | 79.5 | 67.9 | 57.9 | 82.8 | 73.8 | 61.2 | 84.6 | 73.0 |
| 3C-GAN [Li et al., 2020] | ✓ | 57.4 | 79.3 | 81.8 | 69.1 | 79.8 | 80.0 | 66.5 | 56.2 | 83.9 | 72.4 | 60.1 | 85.4 | 72.7 |
| SFADA [He et al., 2024] | ✓ | 57.8 | 78.5 | 82.3 | 68.2 | 79.6 | 79.2 | 66.9 | 57.4 | 83.8 | 73.4 | 58.0 | 84.1 | 72.4 |
| DVD (Ours) | ✓ | **60.1** | 79.6 | **82.5** | 69.1 | **80.8** | 80.6 | 67.9 | 58.5 | 84.3 | 73.5 | 59.3 | **87.6** | **73.7 ± 0.7** |

with a temperature $\tau = 0.13$. We define three parameters: $k_s^{dif}$ and $k_t^{dif}$ (number of neighbors used to parameterize the priors for DVD), and $k_t$ (number of neighbors for preserving the local target feature structure). Unless otherwise stated, we set $(k_s^{dif}, k_t^{dif}, k_t) = (15, 15, 6)$.

**Results.** Tables 1, 2, and 3 present adaptation results on VisDA-C 2017, Office-Home, and Office-31, respectively. We measure performance by classification accuracy (%) on the target domain. "ResNet" denotes the baseline where a pre-trained source model is directly applied to the target domain without adaptation, and "Add." indicates whether a method uses additional modules beyond the backbone during training. As shown, DVD achieves state-of-the-art results on all benchmarks, demonstrating its ability to transfer discriminative knowledge without requiring source data during target adaptation. Recent SFDA studies have also explored ViT-based architectures [Xu et al., 2022, Yang et al., 2023a, Sanyal et al., 2024] as classification backbones. For a fair comparison, we further evaluate DVD with a ViT-B/16 backbone, where it continues to outperform all existing methods (see Appendix C.2).

**t-SNE Visualization.** Figure 2 illustrates the impact of our Discriminative Vicinity Diffusion (DVD) on the VisDA-C 2017 target features. Each point is colored by its ground-truth class label. On the left (before adaptation), domain shift causes heavy overlap across classes, resulting in scattered features and blurred decision boundaries. On the right (after adaptation with DVD), our method first samples from a local Gaussian prior defined by each feature's $k$-nearest neighbors, then applies a drift-only diffusion step to "pull" these samples onto the source manifold. This process propagates label consistency from the source into the target latent space, aligning features with the frozen classifier's decision boundaries. As a re-

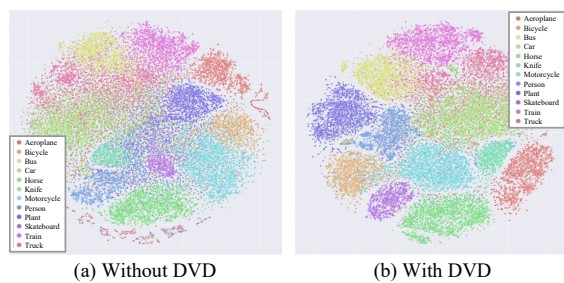

(a) Without DVD    (b) With DVD

Figure 2: t-SNE of target features on VisDA-C 2017.

Table 3: Comparison of SFDA methods on *Office-31* (ResNet-50).

| Method | Add. | A→D | A→W | D→W | D→A | W→D | W→A | Avg. ± s.d. |
|---|---|---|---|---|---|---|---|---|
| ResNet-50 [He et al., 2016] | | 68.9 | 68.4 | 96.7 | 62.5 | 99.3 | 60.7 | 76.1 |
| SHOT [Liang et al., 2020] | | 94.0 | 90.1 | 98.4 | 74.7 | 99.9 | 74.3 | 88.6 |
| AaD [Yang et al., 2022] | | 95.5 | 92.1 | 98.5 | 74.0 | 99.4 | 75.8 | 89.2 |
| NRC++ [Yang et al., 2023b] | | 95.9 | 91.2 | **99.1** | 75.5 | 100.0 | 75.0 | 89.5 |
| SFADA [He et al., 2024] | | 94.8 | 92.0 | 97.6 | 76.5 | 99.8 | 75.7 | 89.4 |
| CPD [Zhou et al., 2024] | | 96.6 | 94.2 | 98.2 | 77.3 | 100.0 | **78.3** | 90.8 |
| 3C-GAN [Li et al., 2020] | ✓ | 93.4 | 92.9 | 97.5 | 76.5 | 99.8 | 77.3 | 89.6 |
| SFADA [He et al., 2024] | ✓ | 95.2 | 91.4 | 98.2 | 77.8 | 100.0 | 76.3 | 89.8 |
| DVD (Ours) | ✓ | **96.7** | **95.2** | 98.6 | **79.2** | **100.0** | 77.4 | **91.2** ± 0.5 |

sult, clusters become tight and well-separated, visually confirming that DVD transfers discriminative knowledge without exposing any source data.

## 4.2 Supervised Single-Domain Classification

Next, we examine how DVD can act as a latent augmentation tool to enhance supervised classification, benefiting source providers. Rather than relying solely on encoder outputs, we generate features via DVD and feed them to the classifier.

**Experimental Setup.** We follow standard training protocols for supervised classification benchmarks. Specifically, we use SGD with momentum $(0.9)$, a weight decay of $5 \times 10^{-4}$, a mini-batch size of $128$, and train for $200$ epochs. The learning rate starts at $0.1$ and follows a cosine annealing schedule [Loshchilov and Hutter, 2017]. DVD is trained with the same hyperparameters used in the SFDA experiments. At test time, for each data, we identify its latent $k$-NNs, generate an augmented feature via DVD, and pass it into the classifier.

**Datasets.** We evaluate DVD-based latent augmentation on three standard benchmarks (additional results on domain adaptation datasets are in the Appendix C.3, demonstrating improved source-domain classification performance):

- **CIFAR-10** and **CIFAR-100** [Krizhevsky et al., 2009]: Each dataset has $60,000$ images ($50,000$ for training, $10,000$ for testing). CIFAR-10 contains 10 classes, and CIFAR-100 has 100 classes.
- **ImageNet** [Russakovsky et al., 2015]: The well-known ILSVRC 2012 benchmark with $\sim 1.2$ million images in $1,000$ object classes.

**Results.** Table 4 presents the top-1 error (%). Although small network architectures are used for illustration (not to set new state-of-the-art), DVD-generated features consistently yield higher accuracy than using the encoder outputs alone. By leveraging $k$-NNs in latent space, DVD effectively enlarges the model's representational power while preserving label consistency. Hence, even with classification parameters held fixed, DVD acts as a latent augmentation module that improves generalization on in-distribution test data.

Table 4: Top-1 test error (%).

| Method | CIFAR-10 | CIFAR-100 | ImageNet |
|---|---|---|---|
| ResNet-18 | 7.07 | 22.74 | 31.46 |
| + DVD | **6.52** ± 0.1 | **22.01** ± 0.2 | **31.06** ± 0.2 |
| ResNet-50 | 6.35 | 22.23 | 24.68 |
| + DVD | **5.92** ± 0.2 | **21.95** ± 0.3 | **24.30** ± 0.3 |
| VGG-16 | 7.36 | 28.82 | 25.82 |
| + DVD | **6.42** ± 0.1 | **26.58** ± 0.3 | **25.02** ± 0.3 |

## 4.3 Domain Generalization

We next demonstrate how DVD enhances *domain generalization*, where models must perform well on unseen target domains *without* any adaptation. Specifically, we consider the source-free domain generalization (SFDG) setting [Cho et al., 2023], which prohibits fine-tuning on target data and focuses on transforming the test data or model outputs.

**Experimental Setup.** We follow [Cho et al., 2023] and adopt ResNet-50, pre-trained with CLIP [Radford et al., 2021], as our backbone. During target inference, we obtain the DVD-generated

feature $\mathbf{z}_1^{t,i}$ and then apply SiLGA leveraging target samples' latent $k$-NN features. For simplicity, we fix $k = 5$ for both the DVD prior parameterization and the SiLGA latent geometry searching.

**Datasets.** We evaluate DVD-generated features on four standard domain generalization benchmarks:

- **PACS** [Li et al., 2017]: 9,991 images across 7 classes, with each domain contributing roughly 2,000 images.
- **VLCS** [Fang et al., 2013]: 10,729 images spanning 5 categories, each domain providing about 2,000 to 3,000 images.
- **Office-Home** [Venkateswara et al., 2017]: The same 65-class dataset used in SFDA, consisting of four distinct domains.
- **DomainNet** [Li et al., 2017]: A large-scale benchmark with 586,575 images across 6 domains and 345 object categories.

**Results.** Table 5 shows the classification accuracy (%) for the target domain. DVD consistently yields strong performance, improving upon ResNet-50 by at least $9\%$ on every dataset. Moreover, it outperforms existing methods without using additional domain-generalization techniques such as momentum ensembling or batch-statistics averaging. This suggests that DVD is complementary to other methods [Lim et al., 2023, Cho et al., 2023] and can be further combined with advanced approaches to achieve even stronger generalization.

Table 5: Domain generalization accuracy (%) using ResNet-50.

| Method | PACS | VLCS | Office-H | DomainNet |
|---|---|---|---|---|
| ResNet-50 [He et al., 2016] | 84.5 | 72.8 | 61.3 | 40.2 |
| ZS-CLIP(C) [Radford et al., 2021] | 90.6 | 76.0 | 68.6 | 45.6 |
| CAD [Dubois et al., 2021] | 90.0 | 81.2 | 70.5 | 45.5 |
| ZS-CLIP(PC) [Radford et al., 2021] | 90.7 | 80.1 | 72.0 | 46.3 |
| PromptStyler [Cho et al., 2023] | 93.2 | **82.3** | 73.6 | 49.5 |
| DVD (Ours) | **93.8**$\pm$ 0.4 | $81.9 \pm 1.2$ | **74.5** $\pm$ 0.8 | **50.8** $\pm$ 0.6 |

### 4.4 Sensitivity to Hyperparameters

DVD has three key hyperparameters: (1) the number of diffusion steps $T$, (2) the source-side neighbor count $k_s^{dif}$, and (3) the target-side neighbor count $k_t^{dif}$. To evaluate each in isolation, we vary one while holding the others at their defaults ($T = 16$, $k_s^{dif} = 15$, $k_t^{dif} = 15$). Figure 3 demonstrates that DVD's performance remains strong and stable across a wide range of settings, confirming that tuning for robust cross-domain classification is straightforward.

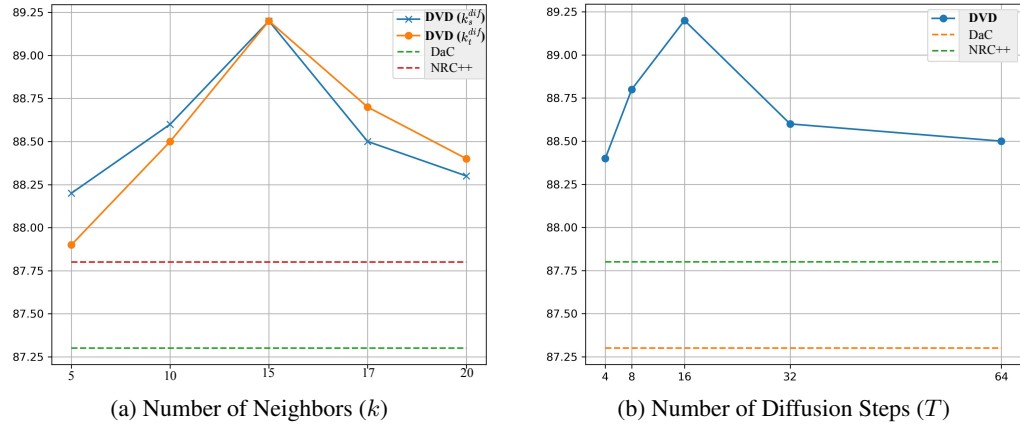

(a) Number of Neighbors ($k$)    (b) Number of Diffusion Steps ($T$)

Figure 3: (**Best viewed in color.**) Hyperparameter sensitivity analysis. The results demonstrate the robustness of our DVD across a range of different hyperparameter settings.

## 4.5   Runtime Analysis

This section quantifies DVD's computational cost during training and at deployment. All timings were collected on an Nvidia V100 GPU and are reported as mean±std over 5 runs using identical conditions. We report:

- **Epoch (s) / Convergence (s)**: training time per epoch and total time to converge under the shared schedule.
- **Inference Time (ms)**: forward-only time from a device-resident input tensor to logits (no host I/O).
- **Latency (ms)**: end-to-end wall-clock time, including minimal host↔device transfers and pre/post-processing.
- **FPS**: throughput under the same setting; approximately FPS ≈ 1000/Inference Time (ms).

**Results.**   Table 6 shows that DVD's per-epoch time (516.3 s) and overall convergence time (5,163.1 s) are on par with NRC++ and SHOT and much faster than DaC. Although DVD trains an auxiliary latent diffusion module, we use only a small number of drift steps, so training overhead remains modest. At deployment, DVD *does not* invoke diffusion; prediction is a single forward pass through $F \circ G_t$, yielding 26.1 ms inference time (38.2 FPS) and 51.1 ms end-to-end latency, effectively matching SHOT and NRC++.

NRC++, SHOT, and DVD all perform a single fixed-weight forward pass at test time. DaC [Zhang et al., 2022] is slower because it maintains a momentum-updated memory bank and computes prototype similarities at inference, adding extra memory access and compute. Since DVD's diffusion is latent-space and training-only, it introduces *no* runtime penalty at deployment.

Table 6: Training cost and deployment efficiency on VisDA-C (ResNet-101, Nvidia V100). All values are reported as mean±standard deviation over 5 runs.

| Method | Epoch (s) | Convergence (s) | Inference (ms) | FPS | Latency (ms) |
|---|---|---|---|---|---|
| DaC [Zhang et al., 2022] | 632.8±3.1 | 12,656.3±55.2 | 50.6±0.8 | 19.8±0.5 | 86.2±1.2 |
| NRC++ [Yang et al., 2023b] | 469.2±2.7 | 4,692.8±41.5 | 26.0±0.6 | 38.4±0.4 | 51.0±0.7 |
| SHOT [Liang et al., 2020] | 439.3±2.4 | 6,589.5±47.2 | 25.9±0.4 | 38.6±0.3 | 50.8±0.6 |
| **DVD (Ours)** | 516.3±2.9 | 5,163.1±43.6 | 26.1±0.5 | 38.2±0.2 | 51.1±0.8 |

## 5   Conclusion

In this paper, we introduced a novel use of LDMs for *explicit* discriminative knowledge transfer, an important yet underexplored capability in settings where raw source data cannot be shared. Motivated by the increasing demand for privacy-preserving and transparent AI, particularly in sensitive domains such as healthcare and finance, we proposed DVD, a method that leverages latent feature geometry to facilitate explicit target adaptation without requiring access to source data. Through $k$-nearest neighbor guidance, DVD enforces label-consistent transformations within the latent space, offering a structured and well-understood mechanism for adaptation under strict privacy constraints. Despite minimal computational overhead, DVD brings value to both sides: target users benefit from reliable and traceable adaptation behavior, while source providers can improve generalization via latent-space augmentation during training. This dual advantage highlights DVD's versatility as a robust and data-privacy-preserving module for explicit cross-domain knowledge transfer.

**Broader Impacts.**   DVD enables privacy-preserving domain adaptation by leveraging LDMs for *explicit* knowledge transfer, addressing a key limitation in existing source-free methods, which often lack clear mechanisms for aligning source and target domains. Its core strength lies in balancing data sensitivity and decision accountability, making it especially suited for high-stakes fields such as healthcare, finance, and scientific research, where both confidentiality and reliable model behavior are significant. By explicitly transferring knowledge without exposing raw source data, DVD reduces the risk of privacy concerns. However, responsible use is important, care must be taken to avoid adapting models to biased or unrepresentative target domains. This highlight the importance of transparent validation and ethical deployment.

## Acknowledgments and Disclosure of Funding

This work was supported by the National Research Foundation of Korea (NRF) grant funded by the Korea government (MSIT) (No. RS-2025-16070597).

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

# Appendix

This appendix provides supplementary materials that expand on the main paper:

- **In-Depth Motivation and Analysis for DVD** (Sec. A): Deterministic diffusion rationale, a graphical binary-classification example, latent-vicinity label propagation, and a probabilistic graphical-model formulation showing how latent vicinities drive cross-domain label transfer.
- **Ablation Studies** (Sec. B): Analyses of the latent-vicinity prior, SiLGA feature aggregation, diffusion *vs.* $k$-NN mean pooling, representational power & generative replay, robustness to vicinity radius $k$ and synthetic domain shifts, effect of diffusion step size $T$, sensitivity to vicinity hyperparameters $(k_s^{\text{dif}}, k_t^{\text{dif}}, k_t)$, and a hyperparameter study (batch size, learning rate, temperature).
- **Additional Experiments** (Sec. C): Fairness of comparisons across SFDA/PPDA/UDA protocols, results with a ViT-B/16 encoder, source-domain feature augmentation, supervised-classification ablations, applicability to OSDA and PDA (DVD-CT), and robustness to initialization seeds.

## A  In-Depth Motivation and Analysis for DVD

We now explain how *Discriminative Vicinity Diffusion* (DVD) transfers knowledge to the target domain without requiring any source data. Our approach extends the classic smoothness assumption from semi-supervised learning [Grandvalet and Bengio, 2004] into the latent spaces of both source and target domains, enabling DVD to effectively bridge domain shifts. This section is structured as follows:

- **Rationale for Deterministic Formulation:** An explanation of why we adopt deterministic latent updates instead of a stochastic variant.
- **Graphical Motivation:** An illustrative binary classification example showing how DVD realigns target features with source decision boundaries by exploiting latent vicinities.
- **Latent Vicinity Label Propagation:** An explanation of how label smoothness within these local neighborhoods supports cross-domain label transfer, aided by a Gaussian prior.
- **Probabilistic Graphical Model Formulation:** A formal treatment of DVD through a latent diffusion framework, highlighting how smoothness propagates across domains via local vicinities.

### A.1  Deterministic Diffusion Rationale

In our DVD, the diffusion module is not used to generate diverse samples, but to reliably transport target features into semantically consistent regions aligned with the frozen source classifier. For this purpose, injecting random noise, which is common in generative diffusion, would be counterproductive. Instead, we design the latent drift to be fully deterministic, so that every generated feature follows a stable and reproducible path. This deterministic choice serves three key aspects of our objective:

- **Privacy and Reliability:** By predicting only the global transport vector $(\mathbf{z}_1 - \mathbf{z}_0)$, the module avoids storing or replaying raw source samples. Each update is reproducible and does not depend on random seeds, ensuring the same target feature is always mapped to the same semantically aligned region.
- **Discriminative Adaptation:** In contrast to generative tasks where diversity is valuable, SFDA requires sharpening class clusters under domain shift. Noise injection can blur decision boundaries and misalign features. A deterministic drift keeps adaptation focused on class-consistent manifolds, preserving discriminability.
- **Efficiency:** Deterministic transport converges in a small number of steps, avoiding error accumulation and unnecessary computation. Stochastic variants require more steps and still degrade accuracy.

Table 7: Ablation on deterministic *vs.* stochastic drift on VisDA-2017.

| Method | Target Acc. (%) | Runtime (second/iter) |
|---|---|---|
| Stochastic Drift (+ Noise) | 87.2 | 2.5 |
| **Deterministic Drift (Ours)** | **88.9** | **1.2** |

To validate this design choice, we compare deterministic drift against a stochastic variant that injects Gaussian noise at each step, using the large-scale VisDA-2017 benchmark. As shown in Table 7, the stochastic version reduces accuracy (–1.7%) and increases runtime significantly. These results confirm that noise injection not only undermines discriminative adaptation but also makes the procedure slower. By contrast, the deterministic scheme consistently yields sharper clusters, stronger alignment, and better efficiency, making it better suited for privacy-preserving domain adaptation.

Overall, the deterministic formulation is not only simpler but also a more principled match to the goal of DVD: to adapt target features efficiently, reproducibly, and in a privacy-preserving way, without compromising semantic alignment.

## A.2 Graphical Motivation

To illustrate the main ideas behind DVD, Figure 4 shows a binary classification example demonstrating how DVD transfers knowledge *without* accessing source data. This process unfolds in two key phases:

- **DVD Training:** We define a prior density by sampling from each source sample's $k$-NN neighborhood in latent space, thereby diffusing its ground-truth label throughout that vicinity. A pre-trained source classifier enforces consistency, ensuring that DVD-generated features align with source labels.
- **DVD Sampling for Target Adaptation:** Once the DVD and source classifier are fixed, only the target encoder is adapted. For each target sample, we form a prior based on its $k$-NN neighbors to generate source-like features. We then apply *contrastive learning* with *SiLGA*, blending these generated features with local target neighbors. This encourages label consistency by realigning target representations to the source decision boundary, effectively bridging domain shifts without the raw source data.

By leveraging local neighborhoods in latent space, DVD explicitly propagates labels from the source domain to the target domain, thus laying the theoretical foundation for cross-domain knowledge transfer.

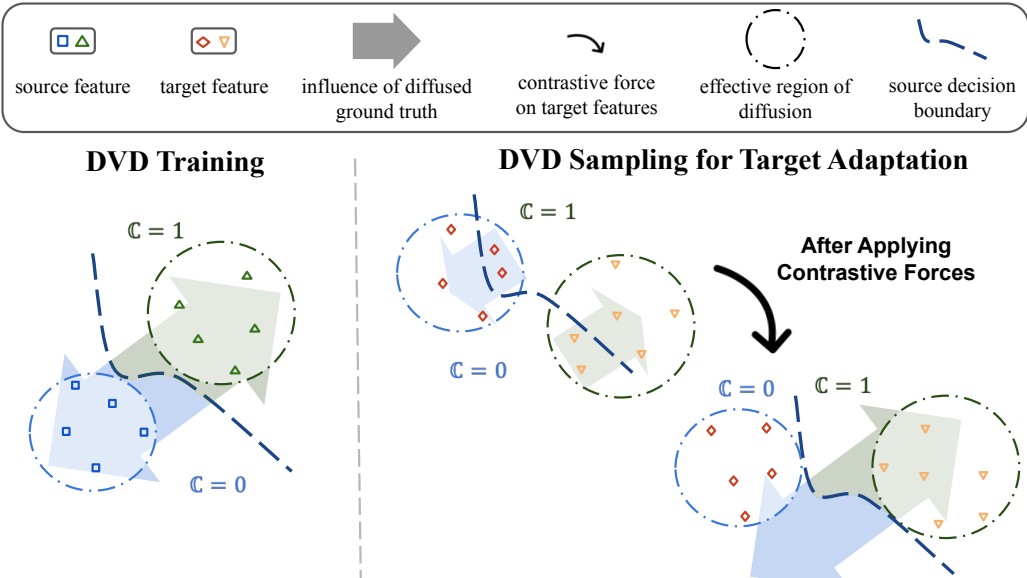

Figure 4: **Graphical Motivation.** A binary classification illustration of DVD-based knowledge transfer without source data (blue: class 0, green: class 1). (1) **DVD Training**: A prior density is defined using the $k$-NN latent vicinity of each source sample, and a pre-trained source classifier "diffuses" ground-truth labels within that neighborhood. (2) **DVD Sampling for Target Adaptation**: With DVD and the classifier frozen, only the target encoder is updated via contrastive learning. We apply *Source-Informed Latent Geometry Aggregation* (SiLGA) to blend DVD-generated features with local target neighbors, thereby aligning target samples to source decision boundaries.

## A.3 Latent Vicinity Label Propagation

DVD builds upon the *latent smoothness assumption*, which states that the nearvy latent feature points are likely to share the same labels [Papernot and McDaniel, 2018]. This idea extends the classical smoothness assumption, which is initially formulated in the input data space [Bezdek et al., 1986], to the latent space of a well-optimized encoder trained with a classification objective. However, both assumptions are confined to scenarios where the data are drawn from the same distribution.

The proposed DVD provides a practical means to extend the latent smoothness assumption to *cross-domain* scenarios, which addresses the challenges posed by domain shifts. This extension is made possible by parameterizing a Gaussian prior using latent vicinities, which are exploited in both domains. Specifically, DVD leverages the similarity of latent vicinities between the source and target domains to propagate labels effectively across domains. The process comprises two key stages:

- **Noise-Adding Training**: During training, the Gaussian prior is parameterized using the mean and variance of the source query's latent vicinity. A neural network is trained to estimate the score function, which enables the diffusion of the source query's label throughout its latent vicinity. The pre-trained source classifier further ensures alignment by matching the DVD-generated feature predictions with the source labels.
- **Noise-Removal Sampling**: In the adaptation phase, the Gaussian prior is parameterized using the latent vicinity of the target query data. This step generates source-like features that share a similar latent vicinity to the target query, which effectively transfers discriminative knowledge to the target domain without requiring access to source data.

Through this mechanism, DVD acts as a *bridge* between the source and target domains, transforming target features into source-like representations by using the underlying latent vicinities of both domains as the guiding keys. For a DVD parameterized by the latent vicinities of two data domains, the following principle applies: "The source label can be encoded into its latent vicinity by a latent diffusion model. Consequently, nearby target features sharing a similar latent vicinity can be transformed by the latent diffusion model into representations likely to share the same label."

## A.4 Probabilistic Graphical Model

To formalize the DVD process, we represent it as a probabilistic graphical model (Figure 5), which captures the stochastic dynamics of latent vicinity diffusion:

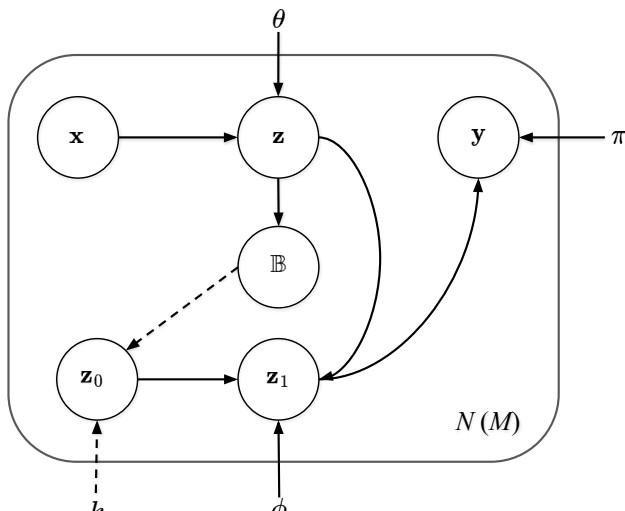

Figure 5: The directed graphical model illustrates how DVD enables explicit knowledge transfer by leveraging *latent vicinity similarities* between the two domains. Solid lines denote the direct causal relations between variables, which include the encoder, the latent diffusion, and the classifier. The dashed lines denote the stochastic sampling from a prior parameterized by the latent vicinity.

**Encoding and Classification.**

- **Encoder**: The encoder $p_\theta(\mathbf{z}|\mathbf{x})$ maps input data $\mathbf{x}$ to latent features $\mathbf{z}$.
- **Classifier**: The classifier $p_\pi(\mathbf{y}|\mathbf{z})$ predicts labels $\mathbf{y}$ based on the latent features $\mathbf{z}$.

**Latent Vicinity Sampling.**

- During **training**, the prior $q_k(\mathbf{z}_0|\mathbb{B})$ initializes the diffusion process using the $k$-NN features of the source latent vicinity.
- During **sampling**, the prior $q_k(\mathbf{z}_0|\mathbb{B})$ is defined by the target latent vicinity.

**Diffusion Process.**   The latent diffusion model $q_\phi(\mathbf{z}_1|\mathbf{z}_0)$ propagates discriminative knowledge across latent vicinities, guided by Gaussian priors.

**Training Objective.**   During training, DVD optimizes the parameters $\phi$ by minimizing the divergence between the source prior and the target posterior:

$$\mathbf{z}_0^s \sim \mathcal{N}(\mu_k^s, \sigma_k^s), \quad \mathbf{z}_1^s \sim q_\phi(\mathbf{z}_1^s|\mathbf{z}_0^s). \tag{12}$$

This process diffuses discriminative knowledge within the source latent vicinity.

**Target Adaptation.**   For target adaptation, DVD leverages the latent vicinity similarity between domains:

$$\mathbf{z}_0^t \sim \mathcal{N}(\mu_k^t, \sigma_k^t), \quad \mathbf{z}_1^t \sim q_\phi(\mathbf{z}_1^t|\mathbf{z}_0^t). \tag{13}$$

This process retrieves discriminative knowledge from the target latent vicinity and generates source-like representations. By progressively aligning target features with source-like representations using contrastive learning, DVD facilitates discriminative knowledge transfer across domains by defining its priors using the latent vicinities from both domains.

**Markov Chain Representation.**   The stochastic sampling process can be expressed as:

$$\mathbf{z}_0^t \to \mathbf{z}_{\alpha_0} \to \cdots \to \mathbf{z}_{\alpha_T} \to \mathbf{z}_1^s. \tag{14}$$

This progression ensures that target features converge to source-like representations supported by label consistency within similar latent vicinities, which enables effective classification in the unlabeled target domain.

By integrating latent vicinity information through Gaussian priors into diffusion processes, as shown in the probabilistic graphical model, DVD offers a theoretically and empirically robust solution to domain shifts during target adaptation without source data.

## B   Ablation Studies

This section dissects the components of DVD through targeted ablation experiments:

- **Latent Vicinity Prior:** Compare our $k$-NN–based Gaussian prior against baseline and noise-based alternatives to validate its impact.
- **Feature Aggregation (SiLGA):** Remove the local-neighbor blending step to assess its role in contrastive alignment.
- **Diffusion *vs.* $k$-NN Mean Pooling:** Replace the diffusion module with simple neighbor averaging to evaluate whether DVD's gains stem from generative transport or from local aggregation alone.
- **Representational Power & Replay:** Contrast DVD on a smaller backbone with a larger vanilla model and disable generative replay to isolate the effect of our diffusion mechanism.
- **Vicinity Radius & Shift Robustness:** Vary $k$ and inject synthetic noise to test the resilience of latent vicinity constructions across domain perturbations.
- **Effect of Diffusion Step Size:** Study how DVD's performance depends on the number of diffusion steps $T$. We vary $T$ at both training and inference to evaluate whether the transport dynamics generalize across mismatched schedules, and we also examine very large $T$, which can cause over-smoothing and degrade discriminative sharpness.
- **$k$-NN Sensitivity:** Vary vicinity hyperparameters $(k_s^{dif}, k_t^{dif}, k_t)$ across datasets to evaluate the robustness to neighborhood size.
- **Hyperparameter Sensitivity:** Examine the effect of batch size, learning rate, and temperature, confirming DVD's robustness across a wide range of values.

## B.1 Impact of the Latent Vicinity Prior

In DVD, we construct the initial diffusion state $\mathbf{z}_0$ by sampling from a Gaussian prior parameterized by the $k$-NNs of each query in latent space. To evaluate the importance of this *latent vicinity prior*, we compare against four alternatives:

- **Baseline**: Directly use the encoder output $\mathbf{z}$ as $\mathbf{z}_0$.
- **Input White Noise**: Inject Gaussian noise into the *input image*, then encode it to form $\mathbf{z}_0$.
- **Latent White Noise**: Inject white noise directly into $\mathbf{z}$ in the latent space and use it as $\mathbf{z}_0$.
- **Centroid Feature Distribution**: Use the mean vector of the latent $k$-NNs ($\mu_0$) as $\mathbf{z}_0$.

Table 8 shows the resulting target-domain classification accuracy (%) on VisDA-C 2017. We observe that simply adding noise at either the input or latent level actually degrades performance compared to the baseline. In contrast, using the latent vicinity to formulate the prior substantially improves accuracy, and our full latent vicinity prior method outperforms all competing strategies. The results highlight the value of leveraging latent vicinities for diffusion.

Table 8: Ablation experiments to evaluate individual DVD module's contributions on *VisDA-C 2017*.

| Method | plane | bcycl | bus | car | horse | knife | mcycl | person | plant | sktbrd | train | truck | Avg. |
|---|---|---|---|---|---|---|---|---|---|---|---|---|---|
| Baseline | 95.8 | 89.3 | 87.4 | 76.8 | 95.1 | 94.2 | 85.6 | 84.7 | 93.2 | 92.5 | 91.3 | 49.8 | 86.3 |
| Input White Noise | 94.5 | 88.5 | 84.2 | 75.3 | 93.6 | 92.4 | 82.7 | 81.8 | 91.5 | 91.9 | 90.2 | 50.8 | 84.8 |
| Latent White Noise | 80.1 | 84.2 | 82.5 | 70.5 | 90.4 | 88.6 | 78.9 | 82.2 | 88.0 | 90.2 | 91.8 | 45.5 | 81.1 |
| Centroid Feature Distribution | 97.4 | 92.4 | **89.6** | 78.2 | 97.7 | 95.8 | 89.8 | **85.4** | 94.9 | 93.2 | 90.4 | 49.6 | 87.9 |
| Without SiLGA | 97.5 | 92.7 | 89.2 | 78.7 | 97.1 | 95.2 | 86.6 | **85.4** | 93.8 | 92.7 | 92.3 | 50.9 | 87.7 |
| **DVD (Full)** | **98.4** | **92.1** | 83.9 | **83.6** | **98.1** | **96.5** | **92.1** | 82.9 | **97.0** | **95.2** | **92.6** | **54.6** | **88.9** |

## B.2 Feature Aggregation for Positive Keys

To mitigate label mismatches between source and target, our method aggregates target $k$-NN features with those from DVD, forming a more informative positive key for contrastive learning. In Table 8, the entry "Without SiLGA" removes this step. The resultant drop in performance confirms that incorporating local target neighbors helps align features more accurately.

## B.3 Diffusion *vs*. Mean Pooling: Isolating the Contribution of DVD

To isolate the contribution of our latent diffusion module, we compare DVD against two simplified variants across three tasks: SFDA (VisDA-2017), supervised classification (CIFAR-10), and domain generalization (DomainNet).

- **Baseline (InfoNCE only):** Standard contrastive learning, where each anchor is paired only with its own data augmentation.
- **$k$-NN Mean Pooling (No diffusion):** For each anchor $z_t$, the $k_t = 6$ nearest neighbors are identified and their mean vector is used as the positive key.
- **DVD (Ours):** Full framework with the latent diffusion module generating positive samples from source-informed priors.

All methods were trained under identical conditions with the same backbone for each dataset.

Table 9: **Ablation study of Diffusion *vs*. $k$-NN Mean Pooling across three tasks.** For VisDA-2017 (SFDA) and DomainNet (DG), higher accuracy indicates better performance ($\uparrow$), while for CIFAR-10 (supervised classification), lower top-1 error is better ($\downarrow$).

| Method | VisDA-2017 (SFDA $\uparrow$) | CIFAR-10 (SC $\downarrow$) | DomainNet (DG $\uparrow$) |
|---|---|---|---|
| Baseline (InfoNCE only) | 82.2 | 6.35 | 40.2 |
| + $k$-NN Mean Pooling (No diffusion) | 86.8 | 6.13 | 46.5 |
| **+ DVD (Ours)** | **88.9** | **5.92** | **50.8** |

**Results.** The results, summarized in Table 9, demonstrate a consistent pattern across all settings:

- On **VisDA-2017 (SFDA)**, nearest-neighbor pooling already provides a significant enhancement (+4.6%) by smoothing noisy target representations. However, DVD adds an additional +2.1% by transporting features toward source-informed clusters, not just averaging within the target domain. This demonstrates the distinct role of our latent diffusion module in cross-domain alignment.
- On **CIFAR-10 supervised classification**, the dataset has no domain shift, yet DVD still reduces error from 6.35% to 5.92%. The gain here is smaller but significant, demonstrating that diffusion sharpens decision boundaries even when the training and testing distributions coincide, confirming its general utility beyond adaptation.
- On **DomainNet (domain generalization)**, the gap is most significant: mean pooling improves accuracy by +6.3%, but DVD delivers a further +4.3% increase. This suggests that under severe distribution shifts, local averaging alone is insufficient, diffusion's generative transport explicitly bridges disparate domains by aligning features to semantically consistent priors.

These results confirm that while neighbor aggregation captures local geometry, it cannot substitute for the latent diffusion. Our DVD consistently outperforms mean pooling by a non-trivial margin, across closed-set SFDA, standard classification, and domain generalization. The consistency of this trend highlights that the latent diffusion module introduces a unique and transferable mechanism: it enforces semantic alignment by pulling features toward class-consistent regions, rather than passively smoothing neighborhoods.

## B.4 Representational Power and Generative Replay

To evaluate whether DVD's performance gains result from effective adaptation rather than increased model capacity, we compare it to a larger source-only ResNet-152 model (60M parameters) on the Office-Home dataset. As shown in Table 10, while larger models often generalize better from source data, DVD – built on a smaller ResNet-50 backbone (40M parameters in total), which achieves 73.7% accuracy-significantly outperforms the ResNet-152 baseline (58.7%). This suggests that the performance gain comes from DVD's adaptation mechanism, not from model size. We also disable the generative replay, which is driven by our latent diffusion module. Without it, performance drops to 69.3%, confirming its importance. Since this replay operates entirely in the target latent space without accessing source data, DVD remains strictly source-free.

Table 10: Comparison of representational power and generative replay on Office-Home.

| Method | Backbone | Accuracy (%) |
| --- | --- | --- |
| Source-only | ResNet-152 (60M) | 58.7 |
| DVD (Ours) | ResNet-50 (40M) | **73.7** |
| DVD w/o replay | ResNet-50 (40M) | 69.3 |

## B.5 Vicinity Radius and Domain Shift Robustness

Our method relies on the assumption that the latent vicinities remain informative across domains, which is motivated by the smoothness principles used in semi-supervised learning, where nearby samples are assumed to share similar labels. In DVD, we extend this across domains using the $k$-nearest neighbors in the latent space to allow the model to synthesize diffusion trajectories guided by localized latent structure.

To evaluate how sensitive DVD is to the choice of vicinity radius ($k$), we vary $k$ from 5 to 25 on Office-Home (Ar→Cl). As shown in Table 11, DVD maintains stable performance across all settings, with accuracy varying by less than 1%. This indicates that the latent vicinity construction is not brittle and does not require fine-tuning of $k$. We further evaluate robustness under synthetic domain shifts by adding Gaussian noise (with standard deviation $\sigma$) to the latent features during adaptation. DVD consistently outperforms SHOT [Liang et al., 2020] across all noise levels, which demonstrates that latent diffusion guided by local structure is more resilient to domain perturbations than purely discriminative alternatives. These results support the claim that latent vicinities generalize well across domains and that the vicinity-guided diffusion mechanism is robust to both hyperparameter changes and representation noise.

Table 11: Robustness of latent vicinity-guided diffusion on Office-Home (Ar→Cl). Left: adaptation accuracy as the vicinity radius $k$ varies; Right: accuracy under increasing synthetic domain noise $\sigma$.

| Method | Vicinity Radius ($k$) | | | | | Domain Noise ($\sigma$) | | | | |
|---|---|---|---|---|---|---|---|---|---|---|
| | 5 | 10 | 15 | 20 | 25 | 0.1 | 0.3 | 0.5 | 1.0 | 1.5 |
| SHOT [Liang et al., 2020] | – | – | – | – | – | 57.1 | 55.4 | 52.8 | 51.9 | 50.3 |
| DVD (Ours) | 58.9 | 59.8 | 60.1 | 59.4 | 58.6 | 60.1 | 59.3 | 57.6 | 56.8 | 55.2 |

## B.6 Effect of Diffusion Step Size

An important design choice in our DVD is the number of diffusion steps $T$. During training, the drift network is optimized to transport features along a discretization schedule of fixed length $T$, where each step corresponds to a specific fraction of the global transport vector. This schedule implicitly encodes how large each update should be and how the drift should behave across intermediate points.

A natural question is whether the model generalizes to a different number of steps at inference, or whether performance degrades when training and inference schedules are mismatched. This ablation is important because in practice one might hope to shorten inference by using fewer steps, or conversely refine adaptation with more steps, without retraining the drift. However, using excessively large $T$ can also be detrimental: each step becomes very small, numerical errors accumulate, and features lose discriminative sharpness. Together, these analyses reveal how tightly the transport dynamics are coupled to the training schedule and how sensitive performance is to step-size choices.

Table 12: Adaptation accuracy on VisDA-2017 under matched and mismatched diffusion steps $T$.

| Train Steps | Inference Steps | Target Accuracy (%) |
|---|---|---|
| 8 | 8 | 88.5 |
| **16** | **16** | **88.9** |
| 32 | 32 | 88.2 |
| 64 | 64 | 87.6 |
| 100 | 100 | 87.1 |
| 100 | 8 | 85.1 |
| 8 | 100 | 86.2 |

Table 12 demonstrates two complementary effects:

- **Matched schedules.** When $T$ at inference matches the schedule used in training, DVD maintains strong accuracy across a wide range of step sizes. The best result is achieved at $T = 16$, while accuracy gradually declines as $T$ grows very large (*e.g.*, 87.1% at $T = 100$). This drop arises because the large step size accumulates numerical errors and cause over-smoothing, where features blur across class boundaries and lose discriminative sharpness.
- **Mismatched schedules.** When $T$ differs between training and inference, performance drops substantially. For example, training with $T = 100$ but testing with $T = 8$ reduces accuracy by 3.8%. This shows that the drift network does not learn a scale-free update rule; instead, it tailors its dynamics to the specific discretization seen during training.

These findings emphasize that the step schedule is an integral part of the transport mechanism learned by DVD. Unlike in generative diffusion where $T$ can often be varied at inference, our discriminative setting requires consistent $T$ to preserve semantic alignment. In practice, this means $T$ should be chosen once, balancing efficiency and accuracy, and kept fixed for both training and deployment.

## B.7 Sensitivity Analysis of $k$-NN Vicinity Selection

To examine the robustness of DVD, we conduct a sensitivity analysis on the vicinity hyperparameters $(k_s^{dif}, k_t^{dif}, k_t)$, which determine how source and target neighborhoods are constructed in the latent space. This study spans three diverse benchmarks: **VisDA-2017**, **Office-Home**, and **DomainNet**, covering synthetic-to-real transfer, multi-domain adaptation, and large-scale heterogeneous shifts.

Table 13: Sensitivity analysis of vicinity parameters across VisDA-2017, Office-Home, and Domain-Net.

| $(k_s^{dif}, k_t^{dif}, k_t)$ | VisDA-2017 (%) | Office-Home (%) | DomainNet (%) |
|---|---|---|---|
| $(5, 5, 3)$ | 87.4 | 72.7 | 49.2 |
| $(10, 10, 5)$ | 88.5 | 73.4 | 49.7 |
| $\mathbf{(15, 15, 6)}$ | **88.9** | **73.8** | **50.8** |
| $(20, 20, 10)$ | 88.3 | 73.1 | 50.0 |
| $(25, 25, 12)$ | 88.0 | 72.8 | 49.6 |
| $(30, 30, 15)$ | 87.8 | 72.6 | 49.5 |
| $(40, 40, 20)$ | 87.5 | 72.3 | 49.1 |

As shown in Table 13, DVD consistently maintains strong accuracy across different settings of the vicinity size on all three benchmarks. When the $k$ values are set too small, the neighborhood fails to capture sufficient local structure, leading to a lack of context and a slight drop in accuracy. On the other hand, very large $k$ values incorporate too many distant or less relevant features, which blur semantic distinctions and weaken class alignment. Between these extremes, moderate settings provide the best balance: they capture enough vicinal information to guide target adaptation, while still preserving the discriminative structure of the latent space.

## B.8 Hyperparameter Study

We report the sensitivity of DVD to three standard hyperparameters: batch size, learning rate, and temperature ($\tau$). Unless otherwise noted, experiments are conducted on *VisDA-C 2017* with ResNet-101 and a fixed default configuration (batch size $= 128$, learning rate $= 3 \times 10^{-3}$, temperature $= 0.13$). These defaults lie within typical ranges used in contrastive SFDA. We used minimal tuning.

Table 14: Ablation on *VisDA-C 2017*: varying one hyperparameter at a time while keeping the others fixed at the defaults (batch size $= 128$, learning rate $= 3 \times 10^{-3}$, temperature $= 0.13$). Reported values are target accuracy (%).

| Batch Size | | Learning Rate | | Temperature | |
|---|---|---|---|---|---|
| Value | Acc. | Value | Acc. | Value | Acc. |
| 32 | 88.2 | $1 \times 10^{-4}$ | 88.1 | 0.05 | 88.2 |
| 64 | 88.6 | $1 \times 10^{-3}$ | 88.6 | 0.07 | 88.5 |
| **128** | **88.9** | $\mathbf{3 \times 10^{-3}}$ | **88.9** | **0.13** | **88.9** |
| 256 | 88.2 | $1 \times 10^{-2}$ | 88.3 | 0.20 | 88.6 |
| 512 | 87.6 | $1 \times 10^{-1}$ | 87.2 | 0.50 | 87.9 |
| | | | | 0.80 | 87.1 |
| | | | | 1.00 | 86.6 |

**Results.** As summarized in Table 14, DVD maintains high accuracy across wide ranges of batch size, learning rate, and temperature, indicating minimal sensitivity and little need for careful tuning. Unlike self-supervised frameworks such as SimCLR that typically rely on very large batches (*e.g.*, 4,096) to obtain sufficient negatives, thereby favoring higher learning rates, DVD (and contrastive SFDA more broadly) pairs each target feature with a source-informed positive, so a moderate batch (*e.g.*, 128) keeps positives influential and supports fine-grained alignment; correspondingly, smaller learning rates stabilize optimization. The temperature $\tau$ controls similarity sharpness: lower values (*e.g.*, 0.13) yield tighter and more discriminative clusters, which are beneficial for adaptation, whereas SSL with massive batches often adopts higher temperatures (0.5–1.0) to avoid collapse. Overall, the flat response across these hyperparameters suggests DVD does not depend on precise and method-specific tuning.

# C  Additional Experiments

Beyond core SFDA evaluations, we extend DVD's validation to diverse settings:

- **Fairness of Comparisons:** Benchmark DVD alongside UDA and PPDA methods under a unified protocol to ensure consistent evaluation.
- **Transformer-Based Encoder:** Evaluate DVD on ViT-B/16 to demonstrate compatibility with modern architectures.
- **Source-Domain Augmentation:** Test DVD-generated features at inference on source domains to measure in-domain gains.
- **Supervised Classification Ablation:** Apply the latent vicinity prior to CIFAR/ImageNet to confirm its benefits beyond domain adaptation.
- **ODA and PSDA Applicability:** Evaluate DVD under open-set and partial-set SFDA settings to demonstrate adaptability beyond the closed-set SFDA protocol.
- **Initialization Robustness:** Run multiple random seeds on Office-Home to compare variance against other SFDA methods.

## C.1  Fairness of Comparisons: SFDA *vs*. UDA *vs*. PPDA

Our setting may appear different from other SFDA methods at first glance, but we emphasize that DVD remains strictly source-free during target adaptation (test time), fully adhering to the core requirement of the SFDA paradigm. During target adaptation, DVD does not access or retain any source-domain data. From a privacy perspective, this means there is no risk of source data leakage at any stage of adaptation. In essence, DVD satisfies the most important requirement of SFDA: **no source data is ever exposed during adaptation**.

Nevertheless, our setting represents a relaxation of the strictest SFDA protocol: while most SFDA methods only transfer a source-pretrained classifier, DVD also trains an auxiliary latent diffusion module offline during the source pre-training. To avoid confusion, we explicitly define DVD as a *privacy-preserving domain adaptation* (PPDA) method in the main paper. To ensure fairness, we extended our empirical evaluation to cover all three settings:

- **SFDA:** No source data or auxiliary modules during adaptation; only the source-pretrained classifier is available.
- **PPDA:** Offline training of auxiliary modules on source data, with zero access to raw source data during target adaptation.
- **UDA:** Full access to labeled source data during adaptation.

Table 15: Comparison on VisDA-2017 with ResNet-101 backbone under different problem settings.

| Method | Setting | Source Access at Adaptation | Target Acc. (%) |
|---|---|---|---|
| CAM+SPLR (A-AT) [Soni and Dutta, 2025] | UDA | ✓ | 69.5 |
| CDTrans [Xu et al., 2022] | UDA | ✓ | 86.9 |
| TTN [Lim et al., 2023] | UDA | ✓ | 85.3 |
| GGF [Zhuang et al., 2024] | UDA | ✓ | 77.6 |
| SFADA [He et al., 2024] | PPDA | ✗ | 83.4 |
| DM-SFDA [Chopra et al., 2024] | PPDA | ✗ | 87.5 |
| **DVD (Ours)** | PPDA | ✗ | **88.9** |

Table 16: Comparison on Office-Home with ResNet-50 backbone under different problem settings.

| Method | Setting | Source Access at Adaptation | Target Acc. (%) |
|---|---|---|---|
| CDAN [Long et al., 2018] | UDA | ✓ | 70.0 |
| TTN [Lim et al., 2023] | UDA | ✓ | 69.7 |
| PGA [Phan et al., 2024] | UDA | ✓ | **75.8** |
| GGF [Zhuang et al., 2024] | UDA | ✓ | 73.6 |
| SFADA [He et al., 2024] | PPDA | ✗ | 72.4 |
| DM-SFDA [Chopra et al., 2024] | PPDA | ✗ | 72.6 |
| **DVD (Ours)** | PPDA | ✗ | 73.7 |

While some existing works with auxiliary modules identify themselves as SFDA, we classify them as PPDA under our stricter protocol. In particular, SFADA [He et al., 2024] and DM-SFDA [Chopra et al., 2024] were originally labeled as SFDA, but both rely on auxiliary modules. We therefore reclassify them as PPDA to more accurately reflect their setting.

As shown in Tables 15 and 16, while DM-SFDA pre-trains its diffusion module on large external datasets, our DVD relies solely on the provided source data yet still achieves higher accuracy (+1.4% on VisDA-2017). This demonstrates that the performance gains of DVD do not stem from additional pre-training or external resources, but from the effectiveness of our vicinal diffusion mechanism. Furthermore, DVD consistently outperforms all PPDA baselines and even matches or surpasses several recent UDA methods that have full access to labeled source data during adaptation. This is particularly significant: despite operating under stricter privacy-preserving constraints, DVD narrows or even closes the gap to conventional UDA. Overall, these results indicate that the PPDA setting does not inherently entail weaker performance, but can in fact deliver competitive or superior outcomes when combined with principled generative modeling. Thus, DVD offers a stronger balance between accuracy and privacy than both SFDA and UDA methods, pointing toward a practical paradigm for real-world deployment where source data sharing is infeasible.

## C.2 Transformer-Based Encoder

To further demonstrate the effectiveness of our DVD across different backbone architectures, we conducted experiments using the ViT-B/16 vision transformer [Xu et al., 2022]. We evaluated DVD on the same SFDA benchmarks as ResNet backbone. The results, summarized in Tables 17, 18, and 19, show that our DVD consistently improves the performance of ViT-based models, which outperform existing SFDA methods built on vision transformers by a large margin. We observe that DVD achieves greater improvements on more effective backbones (larger gains on ViT than on ResNet), which indicates that its knowledge storage and retrieval become more effective with better feature extractors. This is expected, as the quality of extracted features directly impacts the informativeness of the latent vicinity in our DVD.

Table 17: Comparison of SFDA methods on *Office-31* using ViT-B/16.

| Method | A→D | A→W | D→W | D→A | W→D | W→A | Avg. |
|---|---|---|---|---|---|---|---|
| ViT-B/16 | 90.8 | 90.4 | 76.8 | 98.2 | 76.4 | **100.0** | 88.8 |
| CDTrans [Xu et al., 2022] | 97.0 | 96.7 | 81.1 | 99.0 | 81.9 | **100.0** | 92.6 |
| TVT [Yang et al., 2023a] | 96.4 | 96.4 | 84.9 | **99.4** | 86.1 | **100.0** | 93.8 |
| C-SFTrans [Sanyal et al., 2024] | 95.0 | 96.2 | 82.3 | 98.6 | 83.7 | **100.0** | 92.3 |
| DVD (ViT-B/16) | **97.2** ±0.4 | **97.8** ±0.3 | **86.5** ±1.2 | 99.2 ±0.1 | **87.2** ±0.9 | **100.0** ±0.0 | **94.7** ±0.5 |

Table 18: Comparison of SFDA methods on *Office-Home* using ViT-B/16.

| Method | Ar → | | | Cl → | | | Pr → | | | Rw → | | | Avg. |
|---|---|---|---|---|---|---|---|---|---|---|---|---|---|
| | Cl | Pr | Rw | Ar | Pr | Rw | Ar | Cl | Rw | Ar | Cl | Pr | |
| ViT-B/16 | 61.8 | 79.5 | 84.3 | 75.4 | 78.8 | 81.2 | 72.8 | 55.7 | 84.4 | 78.3 | 59.3 | 86.0 | 74.8 |
| CDTrans [Xu et al., 2022] | 68.8 | 85.0 | 86.9 | 81.5 | 87.1 | 87.3 | 79.6 | 63.3 | **88.2** | 82.0 | 66.0 | **90.6** | 80.5 |
| TVT [Yang et al., 2023a] | **73.4** | 84.4 | 86.8 | 81.1 | 85.7 | 86.2 | 78.1 | 70.5 | 87.5 | 83.2 | 73.2 | 88.2 | 82.5 |
| C-SFTrans [Sanyal et al., 2024] | 70.3 | 83.9 | 87.3 | 80.2 | **86.9** | 86.1 | 78.9 | 65.0 | 87.7 | 82.6 | 67.9 | 90.2 | 80.6 |
| DVD (ViT-B/16) | 70.4 | **89.0** | 87.4 | **84.5** | 82.5 | 85.8 | **84.5** | **73.9** | 87.8 | **85.3** | **79.5** | 90.5 | **83.4** |
| | ±0.4 | ±0.5 | ±0.3 | ±1.2 | ±0.5 | ±0.8 | ±0.5 | ±0.6 | ±0.4 | ±0.8 | ±1.2 | ±0.5 | ±0.6 |

Table 19: Comparison of SFDA methods on *VisDA-2017* using ViT-B/16.

| Method | Plane | Bcycl | Bus | Car | Horse | Knife | Mcyle | Person | Plant | Sktbrd | Train | Truck | Avg. |
|---|---|---|---|---|---|---|---|---|---|---|---|---|---|
| ViT-B/16 | **97.7** | 48.1 | 86.6 | 61.6 | 78.1 | 63.4 | 94.7 | 10.3 | 87.7 | 47.7 | 94.4 | 35.5 | 67.1 |
| CDTrans [Xu et al., 2022] | 97.1 | **90.5** | 82.4 | **77.5** | **96.6** | 96.1 | 93.6 | 88.6 | **97.9** | 86.9 | 90.3 | 62.8 | 88.4 |
| TVT [Yang et al., 2023a] | 92.9 | 85.6 | 77.5 | 60.5 | 93.6 | 98.2 | 89.4 | 76.4 | 93.6 | 92.0 | **91.7** | 55.7 | 83.9 |
| C-SFTrans [Sanyal et al., 2024] | 96.2 | 90.1 | 85.0 | 74.2 | 95.0 | 95.7 | 93.1 | 86.9 | 96.8 | 87.3 | 88.7 | 61.5 | 88.3 |
| DVD (ViT-B/16) | 92.5 | **90.5** | **92.7** | **77.5** | 92.5 | **98.5** | **96.0** | **90.0** | 94.5 | **93.3** | 91.0 | **74.5** | **90.2** |
| | ±0.3 | ±0.6 | ±0.5 | ±0.8 | ±0.4 | ±0.5 | ±0.8 | ±0.4 | ±0.2 | ±0.3 | ±0.5 | ±0.9 | ±0.4 |

## C.3 Source-Domain Feature Augmentation

In the main manuscript, we demonstrated how DVD benefits the source provider by enhancing single-domain supervised classification through its role as a latent augmentation module. Here, we conduct additional experiments to evaluate how DVD-generated features directly enhance testing performance on the source domain of domain adaptation datasets. This performance serves as an indicator of the effectiveness of using the latent vicinity and DVD as a repository for ground truth knowledge while addressing data privacy concerns.

**Experimental Setup.** We follow standard training protocols for pre-training on source domains in SFDA. Specifically, we use SGD with momentum $(0.9)$, a weight decay of $5 \times 10^{-4}$, a mini-batch size of 128, and train for 200 epochs. During testing, for each source data, we identify its latent $k$-NNs, generate an augmented feature using DVD, and pass it into the classifier to obtain the prediction score.

**Results.** Table 20 shows that utilizing features sampled from our DVD surprisingly outperforms the standard testing approach, which provides benefits to source model providers. During training, parameterizing the Gaussian prior of DVD with the query's latent vicinity not only encodes its ground truth within that vicinity, but also enriches the feature space with discriminative knowledge. The results again demonstrate that DVD can directly benefit source model providers by acting as a latent augmentation tool.

Table 20: Classification accuracy (%) for source-domain testing using different feature sets. **ResNet** refers to the results obtained by using features encoded directly from ResNet-101.

| Method | Office-31 | | | Office-Home | | | | VisDA-C 2017 |
|---|---|---|---|---|---|---|---|---|
| | A | D | W | Ar | Cl | Pr | Rw | |
| ResNet | 91.5 | 100.0 | 98.7 | 81.5 | 81.5 | 93.9 | 85.1 | 99.6 |
| + DVD | **96.9** | **100.0** | **100.0** | **96.9** | **90.6** | **100.0** | **96.9** | **100.0** |

## C.4 Latent Vicinity Prior Ablation for Supervised Classification

In this section, we present additional ablation experiments on single-domain supervised classification to evaluate the significance of the *latent vicinity prior*. The experiments aim to offer deeper insights and further guidance to the source model provider on effectively using our DVD. Recall that we compare against four alternatives:

- **Baseline**: Directly use the encoder output $\mathbf{z}$ as $\mathbf{z}_0$.
- **Input White Noise**: Inject Gaussian noise into the *input image*, then encode it to form $\mathbf{z}_0$.
- **Latent White Noise**: Inject white noise directly into $\mathbf{z}$ in the latent space and use it as $\mathbf{z}_0$.
- **Centroid Feature Distribution**: Use the mean vector of the latent $k$-NNs ($\mu_0$) as $\mathbf{z}_0$.

Table 21 presents the results of three supervised classification benchmark datasets, using three different backbone networks. Note that **the classification errors** are reported as percentages (with lower values indicating better performance) in this ablation. Similar to the ablation results in VisDA-C 2017, adding noise at either the input or latent level slightly improves testing performance, but ultimately leads to worse performance compared to our baseline. In contrast, formulating the prior using the latent vicinity significantly reduces classification errors, with our full latent vicinity prior method outperforming all other strategies.

## C.5 Applicability of DVD to Open-Set and Partial-Set SFDA

We also explore whether DVD can be extended beyond closed-set SFDA to more challenging protocols such as Open-Set Domain Adaptation (OSDA) and Partial-Set Domain Adaptation (PDA).

**Limitations of DVD.** Our current implementation of DVD explicitly encodes source label consistency into the Gaussian prior for each source class. During adaptation, the drift module generates label-consistent positives, which is ideal for closed-set transfer but not directly suited for open- or partial-set tasks where:

- The target domain may contain classes absent in the source (OSDA).

Table 21: Ablation experiments on supervised learning.

| Method | CIFAR-10 | CIFAR-100 | ImageNet |
|---|---|---|---|
| ResNet-18 | 7.07 | 22.74 | 31.46 |
| Baseline | 6.83 | 22.58 | 31.26 |
| Input White Noise | 6.92 | 22.70 | 31.29 |
| Latent White Noise | 6.94 | 22.54 | 31.34 |
| Centroid Feature Distribution | 6.68 | 22.27 | 31.18 |
| **DVD (Full)** | **6.52** | **22.01** | **31.06** |
| ResNet-50 | 6.35 | 22.23 | 24.68 |
| Baseline | 6.20 | 22.47 | 24.55 |
| Input White Noise | 6.29 | 22.70 | 24.65 |
| Latent White Noise | 6.32 | 22.54 | 24.62 |
| Centroid Feature Distribution | 5.98 | 22.05 | 24.35 |
| **DVD (Full)** | **5.92** | **21.95** | **24.30** |
| VGG-16 | 7.36 | 28.82 | 25.82 |
| Baseline | 7.04 | 27.37 | 25.63 |
| Input White Noise | 7.25 | 27.94 | 25.80 |
| Latent White Noise | 7.18 | 28.05 | 25.52 |
| Centroid Feature Distribution) | 6.71 | 26.96 | 25.16 |
| **DVD (Full)** | **6.42** | **26.58** | **25.02** |

- The source domain may include classes not present in the target (PDA).

In such cases, forcing all target samples to align with source classes risks negative transfer.

**Confidence-Thresholded DVD (DVD-CT).** To extend DVD to OSDA and PDA, we propose a simple yet effective variant: *confidence-thresholded DVD* (DVD-CT). Specifically, after generating source-aligned features $\mathbf{z}^t_{\text{DVD}}$ with the latent diffusion module, we pass them through the frozen source classifier to obtain softmax probabilities $p(\mathbf{y}|\mathbf{z}^t_{\text{DVD}})$. For each target sample, we compute the maximum softmax score $\max_{\mathbf{y}} p(\mathbf{y}|\mathbf{z}^t_{\text{DVD}})$.

Following [Qu et al., 2024], we filter samples using a pre-defined confidence threshold $\tau$:

- In OSDA, samples with confidence below $\tau$ are flagged as belonging to unknown classes, following prior practice in confidence-based rejection methods.
- In PDA, such low-confidence samples are excluded from adaptation to prevent negative transfer from target-private categories.

This mechanism leverages DVD's source-aligned transport to improve the reliability of softmax confidence as a discriminator between known and unknown target classes. Unlike clustering-based OSDA methods [Qu et al., 2023] or subspace-decomposition approaches [Qu et al., 2024], DVD-CT requires no extra modules and can be implemented as a lightweight post-processing step on top of our standard DVD.

Table 22: Comparison with source-free OSDA and PDA methods. For OSDA, results are reported as H-score (%); for PDA, results are top-1 accuracy (%).

| Method | Office-Home (OSDA) | VisDA (OSDA) | Office-Home (PDA) | VisDA (PDA) |
|---|---|---|---|---|
| USD [Jahan and Savakis, 2024] | 61.6 | 57.8 | – | – |
| UMAD [Liang et al., 2021] | 66.4 | 66.8 | 66.3 | 68.5 |
| GLC [Qu et al., 2023] | 69.8 | 72.5 | 72.5 | 76.2 |
| LEAD [Qu et al., 2024] | 67.2 | **74.2** | 73.8 | 75.3 |
| UMAD+LEAD [Qu et al., 2024] | 67.6 | 70.2 | **75.0** | **78.1** |
| GLC+LEAD [Qu et al., 2024] | 70.0 | 73.1 | 73.2 | 75.5 |
| DVD (Closed-set) | 65.6 | 64.2 | 67.5 | 68.0 |
| **DVD-CT (Ours)** | **70.8 (+5.2)** | 73.4 (+9.2) | 73.2 (+5.7) | 75.9 (+7.9) |

**Results.** Table 22 compares DVD and DVD-CT with recent OSDA/PDA baselines. While the original DVD struggles in open/partial scenarios due to its closed-set assumption, DVD-CT achieves competitive improvements, *e.g.*, enhancing VisDA (OSDA) H-score from 64.2% to 73.4%. These

results show that DVD, with minimal modifications, can be extended to OSDA and PDA. While not yet outperforming clustering- or subspace-based methods, DVD-CT demonstrates that source-consistent generative alignment can still contribute to the novel class discovery. In future work, we will explore explicit mechanisms such as adaptive thresholds, subspace decomposition, and unsupervised clustering to further narrow the performance gap.

## C.6 Robustness to Initialization Seeds

We evaluated the robustness of our DVD against random initialization by conducting experiments on the *Office-Home* dataset with five different seeds, replicating the baseline methods with the same seeds for comparison. To illustrate the variance of results across different initializations, we present a bar chart with error bars in Figure 6. The results show that DVD, benefiting from explicit guidance from source-like representations in contrastive clustering, is more robust against random initialization than other SFDA methods.

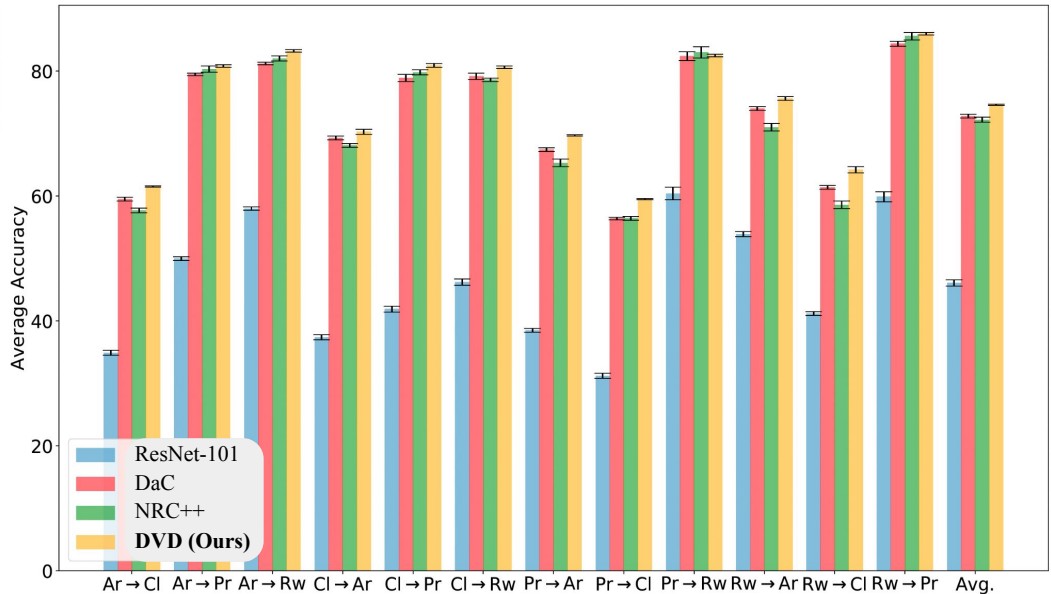

Figure 6: (**Best viewed in color.**) Bar plots with error bars showing classification accuracy (%) across 5 initialization seeds on the *Office-Home* dataset.

