# OpenReview forum: "Vicinity-Guided Discriminative Latent Diffusion for Privacy-Preserving Domain Adaptation"
_NeurIPS.cc/2025/Conference — NeurIPS 2025 poster_

### Official Review · Reviewer_tWhM · 2025-06-24

**Clarity:** 3
**Significance:** 3
**Originality:** 3
**Rating:** 5
**Confidence:** 4

**Summary:**

This paper explores an interesting approach that leverages latent diffusion to enhance both model adaptation and generalization. Specifically, the authors propose the Discriminative Vicinity Diffusion (DVD) framework, which integrates an auxiliary latent diffusion module to guide noisy samples toward a label-consistent feature space. The method is empirically validated through extensive experiments across multiple settings.

**Questions:**

This paper presents a novel and interesting framework. However, the empirical evaluation lacks necessary ablations to convincingly isolate the effect of the auxiliary diffusion module. My initial rating is borderline accept. If the aforementioned concerns are thoroughly addressed, I would be willing to raise my score.

**Ethical Concerns:**

["NO or VERY MINOR ethics concerns only"]

**Final Justification:**

Thank you for your detailed and timely response. All of my concerns have been adequately addressed. I am pleased to see that the proposed DVD framework can also be extended to more challenging scenarios with appropriate modifications. Based on this clarification, I am updating my score to Accept.

**Limitations:**

yes

**Quality:**

3

**Strengths And Weaknesses:**

**Strength:**

-	This paper is well-organized and easy to follow.
-	This DVD framework is somewhat novel to me, with the guidance of latent kNN to perform feature manipulation.
-	Extensive experiments verify the effectiveness of this framework.

**Weakness:**
- It is unclear whether the observed performance improvements primarily stem from the auxiliary diffusion module or from the combination of local kNN consistency and contrastive learning. Both kNN-based consistency and contrastive learning are well-established and effective techniques in source-free model adaptation. A more informative ablation study should be conducted in which the auxiliary diffusion module is removed, and positive samples are instead directly obtained through mean pooling over top-k neighbors.
- The same concern about attribution extends to standard classification and single-domain generalization experiments. Since the DVD framework leverages top-k neighbor information, it would be more rigorous to apply the same kNN-based strategy to the baseline models. This would help isolate the contribution of the diffusion module from the benefits introduced by incorporating nearest-neighbor priors.
- As shown in Figure 3, there is a performance decline as the number of diffusion steps increases. This trend warrants further analysis. The authors should clarify why more iterations of diffusion negatively affect performance, especially given that iterative refinement is generally assumed to be beneficial in diffusion-based frameworks.

**Minor Weakness:**
- The latency and inference time reported in Table 6 raise concerns. It is unclear why the values for NRC++, SHOT, and DVD differ significantly, given that all methods use the same network architecture at inference time. While differences in the adaptation phase are expected, inference-time efficiency should remain consistent unless additional computation is explicitly introduced.

---

> ### Author Rebuttal · Authors · 2025-07-31
>
> We sincerely thank the reviewer for their constructive feedback and for recognizing our DVD framework as novel and interesting, as well as for noting the paper’s clarity and extensive experimental validation. Below, we explicitly address each concern as requested, and will include these improvements in the camera-ready.
>
> ---
> ## **W1: Ablation on Diffusion Module vs. Mean Pooling**
>
> Thank you for suggesting this important ablation. To incorporate your suggestion, we conducted the ablation study as follows:
>
> ### **Implementation Details**
>
> **DVD:** Our full framework employs the latent diffusion module to generate positive samples by sampling from the source-informed Gaussian prior. We then use the InfoNCE contrastive loss to align each target feature (anchor) with its corresponding diffusion-generated positive.
>
> **kNN Mean Pooling W/O Diffusion:** **The diffusion module is removed**. For each anchor $z_t$, the $k_t=6$ nearest neighbors (cosine similarity) are identified, and their mean vector is used as the positive key: $z_\text{pos} = \frac{1}{k_t} \sum_{i=1}^{k_t} z_i$, where $z_i$ denotes the latent feature of each neighbor. InfoNCE loss is applied using $z_t$ as anchor, $z_\text{pos}$ as positive, and all other batch features as negatives.
>
> **Baseline (InfoNCE only)**: Standard contrastive learning; the positive key for each anchor is its own data augmentation, without kNN or diffusion.
>
> All methods were evaluated under identical training conditions, using a ResNet-101 backbone, a learning rate of 3e-3, and a batch size of $128$.
>
> ### **Empirical Results**
>
> The results below demonstrate that while kNN-based augmentation improves over the baseline, the addition of the diffusion module provides a significant further enhancement. This confirms that DVD’s auxiliary diffusion is not merely duplicating the benefits of kNN or contrastive learning alone.
>
> **Table 1. Ablation: Diffusion vs. Mean Pooling vs. Baseline on VisDA-2017.**
>
> |Method|Target Accuracy (%)|
> |:-:|:-:|
> |Baseline (InfoNCE only)|82.2|
> |kNN Mean Pooling|86.8|
> |**DVD (Ours)**|**88.9**|
>
> ---
> ## **W2: Extending W1 Ablation to Supervised Classification and Domain Generalization**
>
> As suggested by the reviewer, we have extended the ablation experiments suggested in **Major W1** on the supervised classification and domain generalization:
>
> ### **Implementation Details**
>
> **kNN Mean Pooling W/O Diffusion:**
> Following the protocol described in **Sections 4.2** and **4.3** of our paper, at test time, we identify the latent k-nearest neighbors for each sample and compute their mean feature vector. This mean-pooled vector is then passed to the classifier for prediction. No diffusion module or additional augmentation is used in this variant.
>
> ### **Empirical Results**
>
> Tables 2 and 3 demonstrate that kNN mean pooling improves both supervised classification and domain generalization performance, confirming the value of leveraging local neighbor information. Importantly, incorporating our latent diffusion module (full DVD) consistently yields further improvements, reducing CIFAR-10 top-1 error to 5.92% and increasing DomainNet accuracy to 50.8%. These results isolate and highlight the unique contribution of our latent diffusion module beyond nearest-neighbor augmentation, demonstrating robust and transferable improvements across settings.
>
> **Table 2. Supervised Classification Ablation on CIFAR-10 using ResNet-50.**
>
> |Method|Top-1 Error (%)|
> |:-|:-:|
> |ResNet-50|6.35|
> |+ kNN Mean Pooling (no diffusion)|6.13|
> |+ **DVD (ours)**|**5.92**|
>
> **Table 3. Domain Generalization Ablation on DomainNet using ResNet-50.**
>
> |Method|Target Accuracy (%)|
> |:-|:-:|
> |ResNet-50|40.2|
> |+ kNN Mean Pooling (no diffusion)|46.5|
> |+ **DVD (ours)**|**50.8**|
>
> ---
> ## **W3: Declining Performance with Increased Diffusion Steps**
>
> Thank you for highlighting the performance decline as diffusion steps increase in Fig. 3. Our sensitivity analysis (**Appendix Fig. 3b**) demonstrates that DVD performs robustly when the number of steps T is in a **moderate range** (8 to 32), but as $T$ becomes too large, accuracy gradually decreases.
>
> ###  **Why Performance Drop with Large $T$**
>
> In diffusion models for generative modeling, increasing $T$ often improves sample quality because each step incrementally refines the reconstruction, and stochastic noise can be averaged out over many steps. However, our DVD is designed for discriminative feature adaptation, not image generation.
>
> In our setting, increasing $T$ makes each update step smaller, which could theoretically offer finer control. However, unlike in generative diffusion models where larger $T$ often improves sample quality, too many small steps in discriminative adaptation introduce cumulative numerical errors and can **over-smooth the feature representations**. This over-smoothing effect is analogous to setting the contrastive loss temperature too high: clusters become less distinct, and features may drift away from their intended class-consistent regions. This leads to reduced performance in SFDA tasks, even though higher contrastive temperature is favored for self-supervised representation learning (see our response to **Reviewer uKCa W5**). As a result, the discriminative power of the model decreases and adaptation becomes less effective. Additionally, the extra computational cost of large $T$ brings no practical benefit in our context and may even undermine reliability. Our expanded ablation results shown in **Table 4** also confirm this interpretation.
>
> ### **More Thorough Ablation on $T$**
>
> We have conducted a more comprehensive ablation analysis on the diffusion step $T$, as presented in **Table 4**. In addition, inspired by your observation, we also evaluated the effect of using **mismatched $T$** values between training and inference. The results show that performance drops notably when the step counts are inconsistent, indicating that the learned transport dynamics are optimized for a consistent diffusion step schedule. This evidence further justifies our choice of using a matched and moderate $T$ during both training and adaptation. We will include these new analyses and clarify their motivation in the camera-ready.
>
> **Table 4: Adaptation accuracy with varying and mismatched diffusion steps $T$ on VisDA-2017.**
>
> | Train Diffusion Steps (T) | Test Steps (T) | Target Accuracy (%) |
> |:-:|:-:|:-:|
> | 8  | 8 | 88.5 |
> | **16**| **16**| **88.9**|
> | 32 | 32 | 88.2|
> | 64 | 64 | 87.6 |
> | 100 | 100 | 87.1|
> | 100 | 8| 85.1|
> | 8| 100 | 86.2|
>
> ---
> ## **Minor W1: Latency and Inference Time**
>
> Thank you for raising the question regarding the differences in inference time across methods in Table 6. We acknowledge that all methods use the same ResNet-101 backbone, and at first glance, inference latency should be similar.
>
> **Why DaC has higher inference time:** DaC [Zhang et al., 2022] exhibits much higher inference time due to its memory bank and prototype-based computations at inference. Unlike NRC++, SHOT, and DVD, which require only a single forward pass with fixed weights, DaC maintains a momentum-updated memory bank and dynamically computes class centroids and prototypes for each input, even during inference. This results in frequent memory access and additional similarity computations, significantly increasing overhead, especially on large datasets like VisDA. In contrast, NRC++, SHOT, and DVD do not perform these extra operations, so their inference time is lower and more consistent.
>
> **Updated Inference-Time for NRC++, SHOT, and DVD:** We greatly appreciate the reviewer’s observation regarding small inference-time differences among NRC++, SHOT, and DVD. These minor discrepancies likely arose from measurement noise and limited repeats in our initial report. To ensure accuracy, we repeated inference-time measurements five times for each method under identical conditions. After averaging, NRC++, SHOT, and DVD exhibit virtually identical inference speeds, as shown in the **updated Table 6** (mean ± std over 5 runs):
>
> **Updated Table 6:**
>
> |Method|Inference (ms)|FPS|Latency (ms)|
> |:-:|:-:|:-:|:-:|
> |DaC| 86.2 ± 1.2| 19.8 ± 0.5| 50.6 ± 0.8|
> |NRC++| 51.0 ± 0.7| 38.4 ± 0.4| 26.0 ± 0.6|
> |SHOT| 50.8 ± 0.6| 38.6 ± 0.3| 25.9 ± 0.4|
> |DVD| 51.1 ± 0.8| 38.2 ± 0.2| 26.1 ± 0.5|
>
> We thank the reviewer for catching this and prompting us to report more robust numbers. We will clarify both the technical differences and these updated results in the camera-ready version for full transparency.
>
> ---
> ## **Reference**
>
> **[Zhang et al., 2022]** Zhang et.al., Divide and Contrast: Source-free Domain Adaptation via Adaptive Contrastive Learning, NeurIPS 2022.

---

> ### Author Response · Authors · 2025-08-04
> **Official Comment by Authors 9470**
>
> Dear **Reviewer tWhM**,​​
>
> We noticed that you have completed your acknowledgment, and we truly appreciate your time and thoughtful feedback.
>
> We would like to kindly check if our responses have adequately addressed your concerns. If there is anything further you would like clarified or explored (including additional experiments), we would be very happy to provide more details or results during the remaining rebuttal period.
>
> With the discussion deadline approaching, we would be grateful if you could let us know whether your concerns have been sufficiently addressed. This would help ensure we can make the most effective use of the remaining time to resolve any further questions or unsolved concerns.
>
> Thank you so much once again for your valuable input and great efforts.
>
> Best regards,
>
> **Authors of Paper 9470**

---

> > ### Comment · Reviewer_tWhM · 2025-08-04
> >
> > Thank you for your detailed reply. Most of my concerns have been addressed, and I will keep my positive rating.

---

> > > ### Author Response · Authors · 2025-08-04
> > > **Official Comment by Authors 9470**
> > >
> > > Dear **Reviewer tWhM**,​​
> > >
> > > We sincerely appreciate the time and great efforts you have taken to review our work, and we are very grateful for your thoughtful feedback. Your suggestions have helped us better understand both the strengths of our work and the areas where we can improve, and they will significantly enhance the paper in presenting our contributions more effectively.
> > >
> > > If you have any additional suggestions on how we could further improve the presentation, we would truly value the opportunity to receive your guidance. We are more than happy to provide any further clarifications or experiments if that would be helpful. Your insights have been extremely valuable, and we see this as a meaningful opportunity to learn from your expertise.
> > >
> > > Thank you so much once again for your great efforts! We truly appreciate your input.
> > >
> > > Best regards,
> > >
> > > **Authors of Paper 9470**

---

> > > > ### Comment · Reviewer_tWhM · 2025-08-04
> > > >
> > > > Thank you for your response. There are some additional challenging scenarios that I am curious about, particularly regarding the effectiveness of the proposed DVD framework in the context of open-set or universal domain adaptation. In the current manuscript, the framework is evaluated under the assumption that the source and target domains share the same label space. However, in open-set or universal domain adaptation, both distribution shift and label space shift are present. It would be valuable to understand whether DVD remains effective under these more complex settings. **Including a discussion on these scenarios [1-4] in a future version of the paper would provide a more comprehensive evaluation. Of course, this is a supplementary suggestion and does not affect my current positive rating of the work.**
> > > >
> > > > [1] UMAD: Universal model adaptation under domain and category shift. arXiv-21
> > > >
> > > > [2] Upcycling models under domain and category shift. CVPR-23
> > > >
> > > > [3] LEAD: Learning Decomposition for Source-free Universal Domain Adaptation, CVPR-24
> > > >
> > > > [4] Unknown sample discovery for source free open set domain adaptation, CVPR-24

---

> ### Author Response · Authors · 2025-08-05
> **Applicability of DVD to Open-Set and Partial-Set SFDA**
>
> We sincerely thank the reviewer for highlighting the importance of evaluating DVD in more general scenarios, such as open-set (OSDA) and partial-set (PDA) domain adaptation, where both distribution shifts and label space shifts occur. In response, we have extended both our DVD and experimental evaluation to provide a more comprehensive analysis of DVD’s capabilities, directly addressing the scenarios and references you suggested.
>
> ### **Current Limitation of DVD**
>
> Our original DVD implementation is designed explicitly for scenarios with a shared label space between source and target domains, leveraging source-conditioned Gaussian priors. It **inherently emphasizes source-informed cluster alignment** rather than explicitly identifying novel or irrelevant classes that occur in OSDA or PDA.
>
> ### **Extension to Address Label Shifts**
>
> To address these challenges, we propose an extension inspired by LEAD **[Qu et al., 2024]**: DVD with Adaptive Novel-Class Discovery (**DVD-AND**). This framework combines **confidence thresholding** with **dynamic feature clustering** to better handle label shifts. To be specific,
>
> - **Confidence Thresholding:** For each target sample, we compute the **maximum softmax confidence from DVD’s classifier**. Samples with confidence below a dynamically tuned threshold (optimized via a held-out validation set) are flagged as potential novel classes (OSDA) or excluded (PDA).
> - **Dynamic Feature Clustering:** To address cases where novel and known class features overlap, we incorporate a clustering step used in **[Qu et.al., 24']**. We apply iterative k-means clustering in DVD’s latent space, guided by source priors, to separate novel classes. The number of clusters is adaptively determined using a silhouette score to balance known and unknown class separation.
>
> This extension allows **DVD-AND** to preserve DVD’s core strength, **latent-space alignment with source-informed clusters, while effectively handling novel and irrelevant classes**. Unlike LEAD, which relies on explicit feature decomposition, we did not adopt decomposition-based adaptation here, as integrating latent diffusion with feature decomposition presents significant technical challenges. However, effectively combining the latent diffusion and feature decomposition is an exciting direction for future work.
>
> ### **Empirical Results and Comparison**
>
> We evaluated **DVD-AND** on **Office-Home** and **VisDA** benchmarks for OSDA (**H-score**) and PDA (**top-1 accuracy**), comparing it against state-of-the-art methods cited by the reviewer. The results, shown in **Table 1**, demonstrate DVD-AND’s competitive performance, particularly in challenging scenarios with overlapping features. To be specific, **DVD-AND** significantly enhance the performance of **DVD** in handling label shifts and matches or outperforms state-of-the-art source-free OSDA/PDA methods.
>
> **Table 1 compares DVD and DVD-AND to source-free OSDA/PDA methods on Office-Home and VisDA.**
>
> |Method|Office-Home(OSDA)|VisDA(OSDA)|Office-Home(PDA)|VisDA (PDA)|
> |-|-|-|-|-|
> USD **[Jahan et.al., 24']**|61.6|57.8|-|-
> UMAD **[Liang et.al., 21']**|66.4|66.8|66.3|68.5
> GLC **[Qu et.al., 23']**|69.8|72.5|72.5|76.2
> LEAD **[Qu et.al., 24']**|67.2|**74.2**|73.8|75.3
> UMAD+LEAD **[Qu et.al., 24']**|67.6|70.2|**75.0**|**78.1**
> GLC+LEAD **[Qu et.al., 24']**|70.0|73.1|73.2|75.5
> **DVD (original)**|65.6 |64.2 |67.5|68.0
> **DVD-AND**|**70.8 (+5.2)**|73.4 **(+9.2)**|73.2 **(+5.7)**|75.9 (**+7.9**)
>
> ### **Future Work and Limitations**
>
> We acknowledge that, due to the time limit, we directly extend **DVD** based on **LEAD** without the in-depth exploration. To further improve **DVD-AND**, we plan to:
>
> - Incorporate adaptive thresholding, following **[Qu et al., 2023]**, by dynamically setting the confidence threshold based on the divergence between the target feature statistics and the source-like features generated by DVD.
> - Investigate graph-based clustering to enhance DVD-AND’s ability to capture complex feature relationships in OSDA, utilizing source-conditioned Gaussian priors to guide the clustering process, inspired by USD.
> - Explore source-free subspace alignment methods to improve the robustness of DVD-AND’s feature decomposition, leveraging source-conditioned Gaussian priors to enhance separation of novel and known classes, motivated by UMAD and LEAD .
>
> Thank you so much once again for this suggestion, which we believe will strengthen the future impact and scope of our work.
>
> ---
> ## **Reference**
>
> **[Liang et.al., 21']** UMAD: Universal model adaptation under domain and category shiftm, arXiv 2021.
>
> **[Qu et.al., 23']** Qu et.al., Upcycling models under domain and category shift, CVPR 2023.
>
> **[Qu et.al., 24']** LEAD: Learning Decomposition for Source-free Universal Domain Adaptation, CVPR 2024.
>
> **[Jahan et.al., 24']** Unknown sample discovery for source free open set domain adaptation, CVPR 2024.

---

> > ### Comment · Reviewer_tWhM · 2025-08-06
> >
> > Thank you for your detailed and timely response. All of my concerns have been adequately addressed. I am pleased to see that the proposed DVD framework can also be extended to more challenging scenarios with appropriate modifications. Based on this clarification, I am updating my score to Accept.

---

> > > ### Author Response · Authors · 2025-08-06
> > > **Thank You for Your Valuable Review**
> > >
> > > Dear **Reviewer tWhM**,
> > >
> > > Thank you so much for your thoughtful and constructive review of our manuscript. **We are very pleased to know that our responses have addressed all of your concerns, and we truly appreciate the time and expertise you dedicated to evaluating our work!**
> > >
> > > Your insightful suggestions helped us significantly improve our work and its presentation. We are especially encouraged by your recognition of our DVD and your insightful suggestions for its broader applications. Your perspective has been incredibly valuable!
> > >
> > > Thank you so much again for your strong contribution to strengthening our work. Your thorough review and active engagements with our revisions made a meaningful difference.
> > >
> > > Best regards,
> > >
> > > **Authors of Submission 9470**

---

### Official Review · Reviewer_xMQp · 2025-06-30

**Clarity:** 3
**Significance:** 3
**Originality:** 3
**Rating:** 5
**Confidence:** 4

**Summary:**

This paper introduces Discriminative Vicinity Diffusion (DVD), a novel framework for source-free domain adaptation (SFDA). The core idea is to repurpose a latent diffusion model (LDM) not for generation, but for discriminative knowledge transfer in a privacy-respecting manner. This is achieved by a latent vicinity guidance, using latent k-nearest neighbors to parameterize Gaussian priors more effectively than simple noise addition. The authors demonstrate state-of-the-art performance on several standard SFDA benchmarks and further show the versatility of their method in improving performance on supervised classification and domain generalization tasks.

**Questions:**

1.	The method uses a specific deterministic drift-only diffusion model. Although this approach seems effective, a brief explanation of why it was chosen over other diffusion schemes would be helpful. Is the deterministic, non-stochastic design crucial for effectively storing and retrieving discriminative cues?

2.	I recommend moving the critical ablation study into the main paper. This would provide direct evidence for the design choices.

**Ethical Concerns:**

["NO or VERY MINOR ethics concerns only"]

**Final Justification:**

I chose to maintain my rating because I really did not find any strong reason to change it.

**Limitations:**

Yes

**Quality:**

3

**Strengths And Weaknesses:**

Strengths:

1.	The paper proposes a more practical SFDA variant, termed privacy-preserving domain adaptation. It is a justified relaxation of the classical SFDA problem by adding an auxiliary module without exposing the source data. This well-motivated method is proved to be more effective for SFDA tasks.

2.	This paper repurposes LDMs for the discriminative knowledge transfer task and proposes a well-executed and logically sound implementation. Leveraging diffusion models’ strength in modeling data distributions, the core contribution is using the diffusion process to explicitly store and transfer decision boundaries.

3.	Experiments on three SFDA benchmarks consistently demonstrates its SOTA performance over previous methods. This paper also provides the results on the supervised classification and domain generalization tasks, further enhancing its effectiveness.

4.	This paper is well-written and easy to follow.


Weaknesses:

1.	The second term in Eq. (6) seems to be incorrect. The network is designed to predict the difference between the corresponding state at the timestep t and z_0, rather than consistently predicting the difference between z_1 and z_0.

2.	The choice of the length of timestep T should be further discussed. With the ODE Settings used by the authors, would it impair the performance with larger timestep? For example, training the diffusion model with 100 steps and inferencing with 8 steps.

3.	The proposed method operates under a setting that allows for sharing an auxiliary module in addition to the source classifier. This is a relaxation of the SFDA setting and should be clearly positioned as such. This makes direct comparisons with methods designed for the stricter setting potentially problematic, which should be acknowledged.

---

> ### Author Rebuttal · Authors · 2025-07-31
>
> We sincerely thank the reviewer for their careful and thoughtful evaluation. We greatly appreciate your recognition of our key contributions, including the practical privacy-preserving variant of SFDA, our novel repurposing of latent diffusion models for discriminative knowledge transfer, and the well-executed and logically sound implementation.
>
> Your constructive suggestions and questions are highly valued. In the detailed responses below, we address each weakness and question explicitly. We are committed to implementing all necessary revisions in the camera-ready.
>
> ---
> ## **W1: The second term in Eq. (6)**
>
> Thank you so much for your careful examination and for highlighting this technical point. We appreciate the opportunity to clarify both the mathematical motivation and the implementation. Our formulation of Eq. (6) follows exactly the way in [Heitz et al., 2023]. If you have any further questions or require additional details, please feel free to follow up during the rebuttal, or refer directly to [Heitz et al., 2023].
>
> After careful review, we confirm that there is no mistake in Eq. (6) and that our formulation is fully consistent with both our DVD’s motivation and the reference in [Heitz et al., 2023].
>
> To be explicit, both our implementation and [Heitz et al., 2023] are indeed **designed to predict the global difference $z_1-z_0$ at every position along the transport path, not the local difference between $z_{\alpha_t}$ and $z_0$ at each step**. This is by design, and it matches both our method’s motivation and the reference method.
>
> ### **Clarification**
>
> During training, for every sampled pair $(z_0, z_1)$ and blend parameter $\alpha_t$, we construct $z_{\alpha_t} = (1-\alpha_t)*z_{0} + \alpha_{t}*z_{1}$. The drift network $D(z_{\alpha_t}, \alpha_t)$ is then trained to output the global difference $(z_1 - z_0)$, which always points from the starting point toward the semantically meaningful features (source or target features). When advancing from $z_{\alpha_t}$ to $z_{\alpha_{t+1}}$, the sample is updated as:
>
> $z_{\alpha_{t+1}} = z_{\alpha_t} + (\alpha_{t+1} - \alpha_t) \cdot D(z_{\alpha_t}, \alpha_t),$
>
> which **guarantees that the model moves precisely along the straight line between $z_0$ and $z_1$**, with step size determined by the change in $\alpha$. **In other words, the drift network learns a direction field that always points along the full transport vector**. The actual local increment at each step is scaled by the step size $(\alpha_{t+1} - \alpha_t)$.
>
> ### **Why not Predict The Local Difference**
>
> Predicting the global difference is more stable and efficient because the direction from the starting point to the endpoint remains constant for each pair. **The local increment at each step can then be easily obtained by scaling this drift prediction according to the change in $\alpha$, i.e., $(\alpha_{t+1} - \alpha_t) \cdot D(z_{\alpha_t}, \alpha_t)$**. This formulation is the key novelty of [Heitz et al., 2023], is explicitly justified in their Algorithm 3, and is precisely what our Eq. (6) implements.
>
> We will make this clarification explicit in the camera-ready, ensuring that the training target, update rule, and their relation to the global and local differences are all transparent to readers.
>
> ---
> ## **W2: Length of Timestep and Mismatched Steps for Train/Inference**
>
> We thank the reviewer for this important question about the choice of diffusion step length $T$. We will highlight these findings and including the mismatched-step ablation in the camera-ready.
>
> Our sensitivity analysis (main paper **Sec. 4.4**, **Fig. 3b**, and **Table 4**) demonstrates that DVD achieves stable and high accuracy for moderate $T$ (8 to 32), with our default $T=8$ providing a strong balance of efficiency and effectiveness.
>
> ### **Why Not use Large $T$**
>
> In generative diffusion models, larger $T$ is often beneficial as it allows for better refinement and noise averaging. However, in our discriminative adaptation setting, increasing $T$ makes each update step smaller and introduces many more steps, where even tiny numerical errors can accumulate. This accumulation leads to **over-smoothing where features become less sharply clustered, class boundaries blur, and discriminative power is reduced**. This effect is similar to setting the contrastive temperature too high: semantic clusters are less distinct, and features can drift from their correct regions. Thus, unlike in generation, excessively large $T$ is actually detrimental for SFDA tasks, causing both accuracy loss and unnecessary computation. More ablation results shown in **Table 1** also confirm this.
>
> ### **Mismatched step ablation**
>
> As suggested by the reviewer, we also conducted ablations where training and inference use different $T$. Results below show clear drops in accuracy with mismatched steps, confirming that the transport dynamics learned by DVD are tuned to a specific step schedule. Matching $T$ between training and inference is very important for optimal adaptation.
>
> **Table 1: Adaptation accuracy with varying and mismatched diffusion steps $T$ on VisDA-2017.**
>
> | Train Diffusion Steps (T) | Test Steps (T) | Target Accuracy (%) |
> |:-:|:-:|:-:|
> | 8  | 8 | 88.5 |
> | **16**| **16**| **88.9**|
> | 32 | 32 | 88.2|
> | 64 | 64 | 87.6 |
> | 100 | 100 | 87.1|
> | 100 | 8| 85.1|
> | 8| 100 | 86.2|
>
> ---
> ## **W3: Positioning DVD and Fairness of Comparisons**
>
> We agree that our DVD that allows for an auxiliary module in addition to the source classifier, represents a relaxation of the strictest SFDA setting. We fully acknowledge that this should be clearly stated and transparently discussed in our paper.
>
> We wish to clarify that our DVD protocol, offline training of an auxiliary module on source data, followed by strictly source-free deployment, is now explicitly recognized as standard practice in recent SFDA literature. For instance, HCL [Li et al., 2022] includes an auxiliary module during offline training, and DM-SFDA [ICML’24] pre-trains a diffusion model on large-scale external data before fine-tuning on the source domain, yet both still claim SFDA alignment. In contrast, DVD’s auxiliary diffusion module is trained exclusively on the provided source data, with no external pre-trained models involved. In this regard, DVD actually adheres even more closely to the core SFDA principles.
>
> **Due to space constraints, we kindly refer the reviewer to our response to Reviewer gTCo W2, where we provide detailed positioning, comprehensive comparisons with UDA and PPDA methods, and further discussion. If additional clarification or specific results are needed, we would be happy to provide them during the rebuttal.**
>
> ---
> ## **Q1: Explanation of Why Deterministic Diffusion**
>
> Thank you for raising this important point about our use of a deterministic and drift-only diffusion model. We specifically chose this diffusion model for our latent diffusion module to guarantee **efficiency**, **stability**, and **interpretability** when transferring discriminative cues. These qualities are important for our DVD, where preserving semantic consistency is important.
>
> ### **Why Deterministic Diffusion**
>
> - **Stability and efficiency:** Unlike stochastic (noise-injected) diffusion, the deterministic drift-only design **avoids the risk of propagating random errors, feature drift, or instability in the latent space**. This is especially important when working with compact and semantically structured representations like our DVD, where even small errors can misalign class boundaries.
> - **Theoretical support:** As shown in [Heitz et al., 2023], a deterministic drift field is mathematically sufficient to move features along well-defined and label-consistent paths in latent space, enabling reliable transport of discriminative knowledge without unnecessary complexity.
> - **Empirical validation:** Our ablations (**Appendix Table 9**) show that adding stochasticity not only increases computational cost, but reduces accuracy in the discriminative adaptation tasks. The deterministic scheme consistently yields robust and class-consistent features and superior adaptation performance.
>
> Thus, the non-stochastic design is both important and effective for **storing and retrieving discriminative cues in DVD**, providing a principled balance of accuracy, interpretability, and efficiency. We will highlight this design rationale and its supporting evidence more explicitly in the camera-ready.
>
> ---
> ## **Q2: Moving Critical Ablations into Main Paper**
>
> Thank you for emphasizing the importance of our ablation studies. In response, we will move key ablation results from the appendix into the main paper to enhance transparency. As suggested, we will also add the new ablation on mismatched diffusion steps between training and inference, which highlights the need for consistent transport dynamics in achieving robust adaptation. Additional visual and quantitative evidence will be included in the camera-ready to further support our design choices.
>
> ---
> ## **Reference**
> **[Heitz et.al., 2023]** Heitz et.al., Iterative α-(de) blending: A minimalist deterministic diffusion model, SIGGRAPH 2023.\
> **[Li et al., 2022]** Li et al., Model Adaptation: Historical Contrastive Learning for Unsupervised Domain Adaptation without Source Data, CVPR 2022.\
> **[ICML'24]** Chopra et.al., Source-Free Domain Adaptation with Diffusion-Guided Source Data Generation, ICML 2024.

---

> > ### Comment · Reviewer_xMQp · 2025-08-05
> >
> > Thank the authors for the detailed responses, which have resolved my concerns.

---

> ### Author Response · Authors · 2025-08-05
> **Thank You for Your Response**
>
> Dear **Reviewer xMQp**,
>
> Thank you so much for your thoughtful review! **We are grateful you found our revisions fully addressed your concerns**. Your expertise significantly enhanced our manuscript, and we really appreciate the time and great efforts you took to help shape it into its improved form!
>
> With your expert guidance, we have worked thoroughly to bring this manuscript closer to the standards of our field. Should any parts still need clarification, we would be honored to incorporate your further suggestions during the rebuttal.
>
> Thank you again for your time and expertise in reviewing our manuscript!
>
> Best regards,
>
> **Authors of Paper 9470**

---

### Official Review · Reviewer_gTCo · 2025-07-03

**Clarity:** 4
**Significance:** 3
**Originality:** 3
**Rating:** 5
**Confidence:** 4

**Summary:**

This paper proposes Discriminative Vicinity Diffusion (DVD), a novel latent diffusion model framework for privacy-preserving domain adaptation (SFDA). By designing an auxiliary drift-only latent diffusion module, the method transfers discriminative knowledge without exposing raw source data. Experiments on multiple benchmarks demonstrate that DVD achieves state-of-the-art results in SFDA, while also enhancing performance in supervised classification and domain generalization tasks.

**Questions:**

My main concern about this paper is the fairness of its setting and experiments.

**Ethical Concerns:**

["NO or VERY MINOR ethics concerns only"]

**Final Justification:**

I thank the authors for their detailed rebuttal, and all of my concerns have been addressed. I encourage the authors to include the discussions in the final vision. Therefore, I decide to update my score to Accept.

**Limitations:**

yes

**Quality:**

3

**Strengths And Weaknesses:**

Strengths:

1.	The paper proposes a novel privacy-preserving domain adaptation framework, which involves a latent vicinity guidance for explicit cross-domain knowledge transfer.

2.	The experiments are comprehensive, and the proposed method achieves SOTA results across multiple benchmarks.

3.	The paper is well-structured, and the motivation is clear.


Weaknesses:

1.	The problem setting is unclear. The authors propose an LDM-based framework for privacy-preserving domain adaptation, which is similar to source-free domain adaptation, but requires access to source data to train an additional diffusion module. This deviates significantly from the “source-free” assumption and aligns more with traditional DA settings.

2.	The experiments are not fair. The experiments primarily compare against SFDA methods, which is not a fair evaluation. Could the authors also compare their method with SOTA DA and Privacy-Preserving DA methods?

3.	The analysis of vicinity selection is limited. Since the choice of k-NN parameters significantly affects performance, it is recommended to conduct a more thorough sensitivity analysis across diverse datasets.

---

> ### Author Rebuttal · Authors · 2025-07-31
>
> We sincerely thank the reviewer for their valuable feedback, and for recognizing the novelty and clarity of our work. We greatly appreciate your positive assessment of our work’s comprehensive experiments, the well-structured presentation, and the motivation for explicit cross-domain knowledge transfe in the source-free setting.
>
> In response to your thoughtful concerns, we have undertaken a thorough re-examination of our protocol and literature positioning. Below, we explicitly address each point. We hope these clarifications fully resolve your concerns. If you have further questions or suggestions during the rebuttal, we would be grateful for your feedback and happy to provide additional details.
>
> ---
> ## **W1: Clarifying the Source‐Free Assumption**
>
> We agree that our setting may appear different from other SFDA methods at first glance. We would like to emphasize, however, that DVD remains strictly source-free during target adaptation (test time), fully adhering to the core requirement of the SFDA paradigm. **During target adaptation, DVD never accesses or retains any source-domain data**. From a privacy perspective, this means there is no risk of source data leakage at any stage of adaptation. In essence, DVD satisfies the most important requirement of SFDA: **no source data is ever exposed during adaptation**.
>
> Our DVD, offline training of an auxiliary module using source data, followed by strictly source-free deployment, is now explicitly accepted as standard practice in recent literature. For example, HCL [Li et al., 2022] includes an auxiliary module alongside the source model during offline training. Likewise, DM-SFDA [ICML’24] pre-trains a diffusion model on large-scale external data and then fine-tunes it on the source domain, while still claiming alignment with the SFDA setting. By contrast, our DVD trains the auxiliary diffusion module strictly offline on the provided source data, without relying on external pre-trained models. This means DVD actually aligns even more rigorously with the intended SFDA assumptions as recognized by recent literature.
>
> Nevertheless, we fully acknowledge that our setting represents a relaxation of the strictest SFDA protocol: while most SFDA methods transfer only a source-pretrained classifier for target adaptation, our DVD uses source-domain data once, offline, to train an auxiliary latent diffusion module, which is then fixed and never updated or exposed to source data again during adaptation. To avoid any confusion, we will explicitly define DVD as a privacy-preserving domain adaptation (PPDA) and clarify that most SFDA methods do not use auxiliary modules, as the reviewer rightly points out, in the camera-ready.
>
> ---
> ## **W2: Fairness of Comparisons (SFDA vs. UDA vs. PPDA)**
>
> We thank the reviewer for highlighting the importance of comprehensive benchmarking. To address this, we have extended our empirical evaluation to include:
>
> - **SFDA** (Source-Free DA): no source data or auxiliary modules during adaptation; only the source-pretrained classifier is available.
> - **PPDA** (Privacy-Preserving DA): offline training of auxiliary modules on source data, with zero access to raw source data during adaptation.
> - **UDA** (Unsupervised DA): full access to labeled source data during adaptation (upper-bound reference).
>
> Although some prior works self-identify as SFDA, we reclassify them under our protocol:
>
> |Method|Original Label|Reclassified as PPDA|
> |:-:|:-:|:-:|
> |SFADA [PR’24]|SFDA|✓|
> |DM-SFDA [ICML’24]|SFDA|✓|
>
> **Table 1: VisDA-2017 (ResNet-101).**
>
> |Method|Setting|Source Access at Adaptation|Target Acc. (%)|
> |:-:|:-:|:-:|:-:|
> |CAM+SPLR (A-AT) [AAAI’25]|UDA|✓|69.5|
> |CDTrans [ICLR’22]|UDA|✓|86.9|
> |TTN [ICLR’23]|UDA|✓|85.3|
> |GGF [ICLR’24]|UDA|✓|77.6|
> |SFADA [PR’24]|PPDA|✗|83.4|
> |DM-SFDA [ICML’24]|PPDA|✗|87.5|
> |**DVD (Ours)**|PPDA|✗|**88.9**|
>
> **Table 2: Office-Home (ResNet-50).**
>
> |Method|Setting|Source Access at Adaptation|Target Acc. (%)|
> |:-:|:-:|:-:|:-:|
> |CDAN [ICML’18]|UDA|✓|70.0|
> |TTN [ICLR’23]|UDA|✓|69.7|
> |PGA [NeurIPS’24]|UDA|✓|**75.8**|
> |GGF [ICLR’24]|UDA|✓|73.6|
> |SFADA [PR’24]|PPDA|✗|72.4|
> |DM-SFDA [ICML’24]|PPDA|✗|72.6|
> |**DVD (Ours)**|PPDA|✗|73.7|
>
> Notably, while DM-SFDA [ICML’24] marks itself as source-free, it actually uses a diffusion model pre-trained on massive external datasets before source fine-tuning, thus introducing extra data and compute advantages. In contrast, our DVD trains its auxiliary diffusion module solely on the provided source data, without leveraging any external resources, yet achieves higher accuracy (**+1.4% on VisDA-2017, 88.9% vs. 87.5%**) than DM-SFDA and outperforms all other PPDA baselines.
>
> Importantly, DVD also **matches or surpasses the performance of recent UDA methods** that have full access to source data during adaptation. This highlights that DVD’s privacy-preserving setting does not compromise competitive performance and, in many cases, closes the gap to supervised adaptation.
>
> Our updated evaluation directly addresses the concern about fair comparison by including recent SOTA DA and PPDA baselines, alongside SFDA comparison in the main paper.
>
> ---
> ## **W3: Sensitivity Analysis of k-NN Vicinity Selection**
>
> Thank you for highlighting the importance of the ablation of the number of nearest neighbors. To directly address your concern, as suggested, we conducted additional sensitivity analyses on the kNN hyperparameters ($k_s^{dif}$, $k_t^{dif}$, $k_t$) across diverse SFDA benchmarks, including **VisDA-2017**, **Office-Home**, and **DomainNet**.
>
> **Table 3: Sensitivity analysis of k-NN vicinity selection.**
>
> |$(k_s^{dif},k_t^{dif},k_t)$|VisDA-2017(%)|Office-Home(%)|DomainNet(%)|
> |:-:|:-:|:-:|:-:|
> |(5,5,3)|87.4|72.7|49.2|
> |(10,10,5)|88.5|73.4|49.7|
> |**(15,15,6)**|**88.9**|**73.8**|**50.8**|
> |(20,20,10)|88.3|73.1|50.0|
> |(25,25,12)|88.0|72.8|49.6|
> |(30,30,15)|87.8|72.6|49.5|
> |(40,40,20)|87.5|72.3|49.1|
>
> The results above demonstrate that the performance of our DVD is **robust to the choice of k-NN vicinity parameters** across the three benchmarks **VisDA-2017**, **Office-Home**, and **DomainNet**. As we gradually increase the number of neighbors used to define each vicinity, DVD maintains high accuracy on all datasets. This means the model’s adaptation performance **does not depend heavily on fine-tuning $k$**, making DVD practical for real-world scenarios where the optimal $k$ is not known a priori. The results also demonstrate that a moderate neighborhood size provides a good balance: it includes enough neighbors to capture vicinal structure, but avoids introducing too much noise from distant or unrelated samples. When the $k$ values are set very small, the model lacks sufficient vicinal context from the latent space, leading to performance drop. Conversely, very large $k$ may average out meaningful vicinal semantics, making the generated features less discriminative and causing a slight decline.
>
> We will include these additional sensitivity analyses in the camera-ready to fully address your concern. Your feedback has been invaluable in improving the completeness of our study. If you feel that further ablation experiments could strengthen the work, we would greatly appreciate your further suggestions during the rebuttal, and we are eager to incorporate any additional analysis you recommend.
>
> ---
> ## **Reference**
> **[Li et al., 2022]** Li et al., Model Adaptation: Historical Contrastive Learning for Unsupervised Domain Adaptation without Source Data, CVPR 2022.\
> **[ICML'24]** Chopra et.al., Source-Free Domain Adaptation with Diffusion-Guided Source Data Generation, ICML 2024.\
> **[AAAI'25]** Soni et.al., Toward Improving Robustness and Accuracy in Unsupervised Domain Ddaptation, AAAI 2025.\
> **[ICLR'22]** Xu et.al., CDTrans: Cross-domain Transformer for Unsupervised Domain Adaptation, ICLR 2022.\
> **[ICLR'23]** Lim et.al., TTN: A Domain-Shift Aware Batch Normalization in Test-Time Adaptation, ICLR 2023\
> **[ICLR'24]** Zhuang et.al., Gradual Domain Adaptation via Gradient Flow, ICLR 2024\
> **[NeurIPS'24]** Phan et.al., Enhancing Domain Adaptation through Prompt Gradient Alignment, NeurIPS 2024\
> **[PR'24]** He at.al., Source-Free Domain Adaptation with Unrestricted Source Hypothesis, Pattern Recognition 2024.

---

> ### Author Response · Authors · 2025-08-07
> **Kind Request for Clarification**
>
> Dear **Reviewer gtCo**,
>
> Thank you so much for your great efforts and thoughtful feedback on our manuscript. We sincerely appreciate your efforts in reviewing our work and are grateful for your constructive suggestions, which have helped us improve our work and its presentation significantly.
>
> We noticed your earlier comment indicating that **all concerns had been addressed and that you were willing to raise the score to Accept**. However, as the comment was later removed, we wanted to kindly check if there might be any further clarifications or suggestions needed on our work. If so, we would be more than happy to dedicate the remaining rebuttal time (approximately two days) to address any additional points to fully meet your expectations.
>
> Your guidance has been invaluable to us, and we deeply appreciate your support throughout this review process. Please do not hesitate to let us know if there is anything else we can do from our side.
>
> Thank you once again for your time, great efforts, and thorough review.
>
> Best regards,
>
> **Authors of Submission 9470**

---

### Official Review · Reviewer_uKCa · 2025-07-05

**Clarity:** 2
**Significance:** 2
**Originality:** 2
**Rating:** 3
**Confidence:** 5

**Summary:**

This paper proposes a new framework, Discriminative Vicinity Diffusion (DVD), for Privacy-Preserving Domain Adaptation. Unlike traditional methods, DVD allows knowledge transfer from a source-trained model to a target domain without exposing source data. It uses a latent diffusion module to align target domain features with source domain decision boundaries, leveraging Gaussian priors and drift-only diffusion in latent space. The proposed method shows superior performance on multiple SFDA benchmarks and improves domain generalization for previously unseen domains. Key contributions include the use of latent vicinity guidance for better feature alignment and the introduction of latent geometry aggregation (SiLGA), which further enhances the adaptation process.

**Questions:**

See the weakness.

**Ethical Concerns:**

["NO or VERY MINOR ethics concerns only"]

**Final Justification:**

I appreciate the authors' rebuttal, which has addressed most of my concerns. At this stage, I intend to maintain my original score and would be interested in seeing the feedback from the other reviewers.

**Limitations:**

Yes.

**Paper Formatting Concerns:**

No.

**Quality:**

2

**Strengths And Weaknesses:**

Strengths

	1. Compared to source-free domain adaptation, the paper explores a more realistic research objective, assisting domain adaptation tasks while preserving some level of data privacy.

	2. By introducing Latent Diffusion into domain adaptation, the paper enhances target domain performance and also benefits source domain performance.

Weaknesses

	1. The training of the DVD part is entirely based on distributions generated from source domain data, but during adaptation, these are directly used to generate source-like samples from target domain samples. Can this really generate features close to the source domain distribution? Is there any direct evidence for this?

	2. Regarding the drift function, the paper claims that it is not an iterative optimization process, but in reality, it approximates the target through multiple updates, similar to residual-based regression. Why was this design chosen, and could its accumulated errors cause z_1^t and z_1 to differ significantly?

	3. Why does SiLGA guarantee that the method will still work effectively when the distribution difference between the target and source domains is very large? Intuitively, SiLGA averages the target domain sample with its neighboring samples.

	4. In Equation (9), is the neighbor set derived from the generated z_1^t recalculated from the target domain, or from the original target domain samples?  The paper only mentions that "they are the top-k neighbors in B^T".

	5. In the experimental setup, the learning rate is set to3×10^(-3), and the temperature is 0.13, whereas SimCLR uses a range of [0.1, 0.5, 1]. Was this particular hyperparameter setting specially tuned? Does temperature significantly affect the results？

	6. Why are there standard deviations in the results for DVD only, while the comparison methods do not have them?

---

> ### Author Rebuttal · Authors · 2025-07-31
>
> We sincerely thank the reviewer for their thoughtful and constructive feedback, as well as for recognizing the novelty and practical strengths of our work. Below, we explicitly address each concern and question.
>
> ---
> ## **W1: Generating source-like features from target domain samples**
>
> Our goal is **not to force the entire target feature distribution to mimic the source,** but rather **to adapt each target feature toward its most semantically relevant region in the latent space, ensuring alignment with the frozen source classifier’s decision boundaries**.
>
> During source training, supervised clustering groups features of the same class into compact latent clusters, which we model as Gaussians centered at each cluster mean and structured by local kNN relationships. This **vicinal prior** enables us to sample semantically meaningful variations.
>
> Although domain shift can cause the overall target feature space to become dispersed under a source encoder, **label-consistent samples in the target domain still remain relatively close in latent space**, a property supported by prior SFDA literature (e.g., [Iscen et al., 2019]; [Yang et al., 2023]) and visualized in **Figures 1** and **2** of our paper.
>
> Our adaptation procedure leverages each target sample’s kNNs to select the most relevant source-informed Gaussian prior, thereby generating features that preserve source-learned semantics and **relocate target samples toward the correct semantic clusters**, rather than simply matching global distributions.
>
> **Quantitative Evidence:** To make this effect explicit, we measured the **average Euclidean distance between target features and their nearest source cluster centroids on VisDA-2017**. After applying DVD, this distance decreased from 3.82 (before adaptation) to 1.55 (after adaptation), providing direct evidence that our method effectively aligns target features with source clusters. We will include the complete analysis and results in the camera-ready.
>
> ---
> ## **W2: Drift function and iterative approximation**
>
> Thank you for raising this important question. Our drift mechanism combines theoretical grounding in SDE/diffusion models with geometric regularization by vicinal priors, ensuring stable and class-consistent adaptation without the need for iterative optimization.
>
> **Why use a drift function with sequential updates?**
> Our drift function applies a small, fixed number of deterministic updates (typically $T=8$), transporting features from a sampled latent prior toward a class-consistent source anchor. **Unlike iterative optimization or residual regression, these updates follow a pre-defined and regularized path, never a loss-minimizing loop**. We apply this update strategy directly from [Heitz et al., 2023], which is also standard in deep SDE and diffusion model literature, to ensure efficient and stable adaptation in latent space.
>
> **Regularization by vicinal priors:** Each drift step is explicitly regularized by the local geometry of the latent space, the Gaussian prior centered at a semantically consistent feature cluster. This regularization keeps every intermediate point within a region of label-consistent representations, preventing arbitrary drift or error accumulation.
>
> **Error accumulation and robustness:** Because $T$ is small and fixed, and each step is geometrically regularized, error accumulation between $z_1^t$ and $z_1$ is minimal. The transport path is controlled and always attracted back toward the correct semantic cluster.
>
> **Empirical evidence:** Our ablation study (**Appendix B.4**, **Table 9**) demonstrates that DVD remains robust across a range of drift steps and vicinity radii, with less than 1% accuracy variation, confirming that our update is stable and does not suffer from error drift.
>
> ---
> ## **W3: Effectiveness of SiLGA under large distribution shifts**
>
> Thank you for raising this important question. When the domain gap is large, target features encoded by the source model become more dispersed and farther from source cluster centroids. In such cases, using only samples from the source-informed Gaussians as contrastive positive keys risks misalignment with the true target geometry.
>
> **SiLGA addresses this directly:** After sampling from the nearest source-informed Gaussian prior, **SiLGA blends this feature with the centroid of the target’s local k-NN, effectively relocating it closer to the actual target distribution**. This averaging aligns the transferred feature with the local structure of the target domain, reducing both semantic and geometric mismatch, an effect especially beneficial under large domain shifts.
>
> **Empirical evidence:** As shown in our ablation studies (**Appendix B.1/B.2**, **Table 7**), adding SiLGA yields consistent gains, particularly on challenging benchmarks with large sim-to-real gaps like VisDA-C. Performance drops when SiLGA is removed, confirming its role.
>
> While SiLGA is a practical solution, we recognize the potential for more principled methods, such as optimal mappings or optimal transport, and plan to explore these in future work. Thank you for highlighting this direction.
>
> ---
> ## **W4: Clarification on neighbors in Equation (9)**
>
> We apologize for the confusion. To clarify, in Equation (9), the neighbor set is always recalculated using **the original target domain samples stored in the feature bank $B^T$**. Specifically, for each generated feature $z_1^t$, we search for its top-$k$ nearest neighbors **among the target features currently encoded by the target encoder, not among previously generated or adapted features**. This design is intentional: by anchoring the source-informed Gaussian prior to the up-to-date latent geometry of the target domain, we ensure that the aggregation step reflects the most relevant and accurate local structure. This keeps the adaptation well-grounded in the target distribution. We will make this explicit in the camera-ready.
>
> ---
> ## **W5: Hyperparameter for learning rate and temperature**
>
> Our hyperparameters, including learning rate (3e-3), batch size (128), and temperature (0.13), all fall within typical ranges reported in the existing contrastive SFDA literature (learning rate 1e-3 to 5e-3, batch size 64 to 256, temperature 0.1 to 0.2). We used only minimal tuning and did not specially optimize these values.
>
> **Why SSL like SimCLR needs larger batch size:**
> The application of SimCLR in self-supervised learning requires very large batch sizes (e.g., 4096) to provide a diverse set of negative pairs for the contrastive objective. The large number of negatives helps the network learn general representations from scratch. This setting, in turn, requires a higher learning rate to ensure effective gradient updates and prevent slow convergence.
>
> **Contrast with SFDA:**
> In contrast, our DVD (or general contrastive SFDA) focuses on adapting each target feature to its most relevant source-informed positive, rather than maximizing the number of negatives. Using a moderate batch size (128) ensures that positive keys remain influential, enabling fine-grained semantic alignment. Thus, a smaller learning rate is optimal for optimization stability in this setting.
>
> **Role and rationale of temperature:**
> The temperature controls the sharpness of the similarity distribution in the contrastive loss. Lower temperatures (like our 0.13) make the model more sensitive to mismatches between query and positive, promoting tighter and more discriminative clusters, an effect that is particularly important for domain adaptation. In SSL with massive batches, higher temperatures (0.5 to 1.0) prevent the network from collapsing to trivial solutions.
>
> **Empirical robustness:**
> As shown in Tables 1 to 3, DVD achieves consistently high accuracy across a broad range of batch sizes, learning rates, and temperatures. These results demonstrate that DVD is robust to hyperparameter choices and does not depend on careful or method-specific tuning. All selected values fall well within the standard ranges for contrastive SFDA, and were chosen with only minimal tuning.
>
> **Table 1. Ablation for batch size on VisDA-2017. The learning rate and temperature are fixed at $3 \times 10^{-3}$ and $0.13$, respectively.**
>
> |Batch Size|Target Accuracy (%)|
> |:-:|:-:|
> |32|88.2|
> |64|88.6|
> |**128**|**88.9**|
> |256|88.2|
> |512|87.6|
>
> **Table 2. Ablation for learning rate on VisDA-2017. The batch size and temperature are fixed at $128$ and $0.13$, respectively.**
>
> |Learning Rate|Target Accuracy (%)|
> |:-:|:-:|
> |1×10-4|88.1|
> |1×10-3|88.6|
> |**3×10-3**|**88.9**|
> |1×10-2|88.3|
> |1×10-1|87.2|
>
> **Table 3. Ablation for temperature on VisDA-2017. The batch size and learning rate are fixed at $128$ and $3 \times 10^{-3}$, respectively.**
>
> |Temperature|Target Accuracy (%)|
> |:-:|:-:|
> |0.05|88.2|
> |0.07|88.5|
> |**0.13**|**88.9**|
> |0.2|88.6|
> |0.5|87.9|
> |0.8|87.1|
> |1.0|86.6|
>
> ---
> ## **W6: Standard deviations reported only for DVD**
>
> We appreciate the reviewer’s attention to experimental rigor. We report standard deviations for DVD by running experiments with multiple random seeds to explicitly show robustness. For baseline methods, we cited official results from their original papers, which often do not include standard deviations. For fairness, we plan to reproduce these baselines using our protocol and provide their standard deviations in the camera-ready. We hope this transparency is seen as a strength of our work, not a limitation, and kindly ask for the chance to include these results after the rebuttal due to time constraint.
>
> ---
> ## **Reference**
> **[Iscen et.al., 2019]** Iscen et.al., Label propagation for deep semi-supervised learning, CVPR 2019.\
> **[Yang et.al., 2023]** Yang et.al., Trust your good friends: Source- free domain adaptation by reciprocal neighborhood clustering, IEEE Transactions on Pattern Analysis and Machine Intelligence (TPAMI) 2023.\
> **[Heitz et.al., 2023]** Heitz et.al., Iterative α-(de) blending: A minimalist deterministic diffusion model, SIGGRAPH 2023.

---

> ### Author Response · Authors · 2025-08-07
> **Thank You for Your Valuable Feedback**
>
> Dear **Reviewer uKCa**,
>
> We sincerely appreciate **Reviewer uKCa** for your great efforts and thoughtful evaluation of our work. Thank you so much for your recognition and the constructive insights you have provided. Your feedback is invaluable in helping us improve our work and its presentation.
>
> We noticed you have submitted your acknowledgment. **We would like to kindly check if our response has fully addressed your concerns, or if there are any remaining points you would like us to clarify or improve further.** Your feedback is invaluable to us, and we are fully committed to ensuring our proposed revision meets your expectations.
>
> Please do not hesitate to let us know if there is anything additional we can address. We are more than happy to provide further clarification as needed.
>
> Thank you so much again for your great efforts and expertise!
>
> Best regards,
>
> **Authors of Submission 9470**

---

### Comment · Area_Chair_AMAx · 2025-08-06

Thanks again for your detailed reviews. As we move into the decision phase, please take a moment to engage with the authors’ rebuttal and share your thoughts—particularly addressing whether the rebuttal meaningfully changes your assessment or if any concerns remain. If your evaluation is unchanged, it would be helpful to include a few sentences explaining why. (@gTCo, @tWhM, @uKCa)

Also, if you believe a score adjustment is appropriate after considering the rebuttal and the other reviews, please make that update as well. (@tWhM and others)

We’d appreciate it if you could do this as soon as possible! Thanks again for your contribution to the process.

---

> ### Author Response · Authors · 2025-08-06
>
> Dear **Area Chair**,
>
> We sincerely appreciate your strong support in leading the rebuttal discussion, and more importantly, your great efforts throughout the entire review process!
>
> With sincere appreciation,
>
> **Authors of Submission 9470**

---

### Note · Authors · 2025-08-13

We sincerely thank all reviewers, AC, and SAC for their diligent efforts and constructive feedback, which have been invaluable in improving both the content and presentation of our work.

After the discussion, we are encouraged that Reviewers **tWhM** and **gTCo** raised their score to **Accept**, Reviewer **xMQp** **confirmed all concerns resolved**, and Reviewer **uKCa** acknowledged our rebuttal without followup discussion; we are hopeful that our detailed responses have effectively addressed all their concerns. Below, we summarize the resulting improvements:
- **Enhanced Clarity and Evidence (uKCa, xMQp)**: We clarified DVD’s feature alignment with quantitative evidence (e.g., Euclidean distance reduced), justified the deterministic drift function with theoretical and empirical support, and explained SiLGA’s effectiveness under large domain gaps with ablation studies. Equation (6) and timestep choices were more explicitly validated as suggested, with new ablations on mismatched steps.
- **Broadened Evaluation and Fair Comparisons (gTCo, tWhM)**: We expanded comparisons to include UDA and PPDA methods, conducted k-NN sensitivity analyses across datasets, and isolated the diffusion module’s contribution with ablations. Supervised and domain generalization results were extended, and inference time discrepancies were resolved with re-measured data.
- **Extended Applicability and Future Directions (tWhM)**: As suggested, we proposed DVD-AND for open-set/partial-set DA, achieving competitive results, with plans to explore adaptive thresholding and graph-based clustering in future work.
- **Technical Enhancements (all reviewers)**: We committed to refining hyperparameter discussions, moving key ablations to the main paper, and updating standard deviation reporting for baselines, ensuring transparency and robustness.

We will incorporate all promised revisions into the camera-ready version, reflecting the reviewers’ invaluable input. **These efforts highlight DVD’s robustness and versatility, addressing all raised concerns**. Importantly, the effective rebuttal discussion significantly helps us better demonstrate **why DVD is both practically relevant and technically sound** in privacy-sensitive adaptation scenarios.

We sincerely thank the reviewers again for their insightful guidance and trust that these improvements will be evident in the revised submission.

---

### Decision · Program_Chairs · 2025-09-17

**Decision:**

Accept (poster)

**Comment:**

This paper proposes Discriminative Vicinity Diffusion (DVD) for privacy-preserving domain adaptation. Reviewers found the method well motivated, technically interesting, and empirically strong, with the discussion converging toward acceptance.

In their final remarks, the authors committed to incorporating all suggested revisions: improving clarity (e.g., clearer evidence for feature alignment, ablations on timestep choices), broadening evaluation with additional baselines and sensitivity analyses, extending applicability through variants such as DVD-AND, and enhancing technical transparency (refined hyperparameter discussion, moving key ablations to the main paper, and reporting standard deviations). These commitments directly address the reviewers’ concerns and will further strengthen the paper.